# Loss of LCMT1 and biased protein phosphatase 2A heterotrimerization drive prostate cancer progression and therapy resistance

Loss of the tumor suppressive activity of the protein phosphatase 2A (PP2A) is associated with cancer, but the underlying molecular mechanisms are unclear. PP2A holoenzyme comprises a heterodimeric core, a scaffolding A subunit and a catalytic C subunit, and one of over 20 distinct substrate-directing regulatory B subunits. Methylation of the C subunit regulates PP2A heterotrimerization, affecting B subunit binding and substrate specificity. Here, we report that the leucine carboxy methyltransferase (LCMT1), which methylates the L309 residue of the C subunit, acts as a suppressor of androgen receptor (AR) addicted prostate cancer (PCa). Decreased methyl-PP2A-C levels in prostate tumors is associated with biochemical recurrence and metastasis. Silencing LCMT1 increases AR activity and promotes castration-resistant prostate cancer growth. LCMT1-dependent methyl-sensitive AB56αCme heterotrimers target AR and its critical coactivator MED1 for dephosphorylation, resulting in the eviction of the AR-MED1 complex from chromatin and loss of target gene expression. Mechanistically, LCMT1 is regulated by S6K1-mediated phosphorylation-induced degradation requiring the β-TRCP, leading to acquired resistance to anti-androgens. Finally, feedforward stabilization of LCMT1 by small molecule activator of phosphatase (SMAP) results in attenuation of AR-signaling and tumor growth inhibition in anti-androgen refractory PCa. These findings highlight methyl-PP2A-C as a prognostic marker and that the loss of LCMT1 is a major determinant in AR-addicted PCa, suggesting therapeutic potential for AR degraders or PP2A modulators in prostate cancer treatment.

Reversible protein phosphorylation plays a fundamental role in numerous biological processes and is regulated by a dynamic interplay between protein kinases and phosphatases. Disruption of this balance due to aberrant activation of kinases and inactivation of phosphatases is a hallmark of cancer[1]. Though the inappropriate activation of numerous oncogenic kinases plays a causal role in transformation, metastasis, and therapy resistance and several of these kinases have been successfully targeted for cancer treatment, much less is known about the function and mechanisms of regulation of phosphatases in cancer. The highly conserved protein phosphatase 2A (PP2A) is a major serine/threonine phosphatase with a complex heterotrimeric composition involved in the regulation of numerous critical aspects of cellular function[2]. PP2A is one of the most abundant enzymes, accounting for up to 1% of the total cellular protein in some tissues and represents the majority of serine/threonine phosphatase activities in many tissues[2,3]. PP2A is comprised of a core dimer consisting of a scaffolding "A"

✉ e-mail: egon.ogris@meduniwien.ac.at; gnarla@med.umich.edu; asangani@upenn.edu

subunit (PPP2R1/PP2A-A), and a catalytic "C" subunit (PPP2C/PP2A-C), which form a heterotrimeric complex with one of at least 20 substrate-directing regulatory "B" subunits clustered into four structurally distinct families PPP2R5/B56, PPP2R2/B55, PPP2R3/PR72/PR130, and Striatin[3,4]. The AC heterodimers are not considered the functional phosphatase holoenzyme in vivo, with the binding of the B subunit conferring target specificity and allowing PP2A to dephosphorylate target proteins.

PP2A is widely considered a tumor suppressor and plays a fundamental role in regulating cellular homeostasis by dephosphorylating a broad array of protein targets, from mitotic spindle components, chromatin regulators to intracellular signaling mediators[2,5,6]. PP2A dysregulation is commonly observed in cancer, typically through one of several mechanisms, including somatic mutations, haploinsufficiency and/or decreased expression of PP2A subunits, and increased expression of endogenous PP2A inhibitors such as CIP2A[7–10]. PP2A heterotrimerization is regulated through post-translational modifications of the catalytic C subunit, which influence the binding affinity of the B subunits to the AC dimer and thus the substrate specificity. While PR130 and Striatin binds to the AC dimer in a methylation-independent manner, a central evolutionarily conserved mechanism that leads to heterotrimerization of AC and B55, and B56 subunits is the S-adenosylmethionine dependent PP2A-C α-carboxymethylation at the terminal leucine residue (L309) by Leucine Carboxyl Methyltransferase 1 (LCMT1)[2,11–14]. The presence of methylation on L309 neutralizes the negative charge on the terminal carboxylic acid, allowing for the binding of both B55 and B56 family members to the PP2A AC dimer. Conversely, Protein Phosphatase Methylesterase 1 (PME-1) mediated removal of this methylester from the PP2A-C dimer results in the retention of the negatively charged terminal carboxylic acid creating steric hinderance that prevent B55 and B56 family member binding, thus adding another layer to the regulation of PP2A biogenesis[2,15,16]. Given the previous studies demonstrating the role of the B55/B56 methyl-sensitive PP2A heterotrimers in regulating gene expression by targeting multiple signaling pathways such as PI3K/AKT/mTOR activation[17–20], one critical function of C subunit methylation by LCMT1 could be to coordinately regulate both B55/B56 containing heterotrimers to negatively affect transcription and survival signals in cancer. Interestingly, LCMT1 is the only enzyme known to catalyze methylation of not only PP2A-C but also two other PPP family members, the PP4 and PP6 catalytic subunits[21,22]. Nevertheless, the status of LCMT1 expression, regulation, and its role in cancer development and progression remains elusive.

Prostate cancer (PCa) is the second most common cancer in men worldwide, and the eighth leading cause of cancer-related deaths[23]. PCa relies on AR driven transcription for growth and survival, forming the basis for effective androgen deprivation therapy (ADT) as a first-line of treatment in the primary stages of the disease. However, restoration of AR signaling through AR point mutations, AR overexpression, AR genomic amplification, and constitutively active AR splice variants contributes to an AR-addicted castration-resistant prostate cancer (CRPC)[24–26]. Next-generation AR signaling inhibitors (ARSIs) such as abiraterone, enzalutamide, darolutamide, or apalutamide are used in combination with ADT to treat metastatic and non-metastatic CRPC[27,28]. Despite the clinical success surrounding ARSI, resistance to these therapies invariably emerges through various mechanisms that restore AR-signaling, resulting in poor clinical outcomes[29–31]. AR-mediated transcriptional addiction is, therefore, a hallmark of a vast majority of refractory CRPC, where AR functions in a multi-protein complex comprising transcriptional coactivators and chromatin-associated proteins[29,32,33]. Additionally, greater than 50% of CRPC demonstrates PI3K pathway activation through either biallelic loss of *Pten* or activating mutations in PIK3CA and AKT1[24,34]. Nevertheless, how this recurrently activated oncogenic survival pathway converges upon AR signaling is unclear. There remains an impending need for a better understanding of the mechanism that restores AR signaling and to discover novel ways to target AR addiction in refractory CRPC.

In this study, we report the association of decreased PP2A-C α-carboxymethylation of L309 with rapid biochemical recurrence and metastasis in PCa. Our results show that reduced AB56αC heterotrimers lead to increased AR activity and castration-resistant tumor growth in vivo. Furthermore, we demonstrate LCMT1-dependent methyl-sensitive AB56αC heterotrimer targets AR and its critical coactivator MED1 for dephosphorylation, resulting in the recruitment of the AR-MED1 complex from the chromatin and loss of target gene expression. In addition, we identified S6K1/β-TRCP mediated degradation of LCMT1 as a mechanism for the restoration of AR signaling in antiandrogen refractory prostate cancer cells. Together, these results demonstrate that the loss of LCMT1/decreased PP2A-C α-carboxymethylation is a major determinant of AR-addicted PCa and supports the therapeutic use of AR degraders and selective small molecule modulators of PP2A for refractory prostate cancer treatment.

## Results

### Methyl-PP2A-C loss is associated with prostate cancer progression

LCMT1 catalyzed PP2A-C α-carboxymethylation at L309 is required for the assembly of B55 and B56 containing holoenzymes that negatively regulate signaling cascades including the PI3K-AKT-mTOR pathway (Fig. 1a). The extent to which L309 methylation drives the assembly of each type of specific B56 containing holoenzyme has not been fully elucidated. In contrast to the B56 holoenzyme family, where the role of methylation is equivocal, the B55 holoenzyme exhibits an obligate requirement for this modification[13,14,21,22]. To better define the role of carboxymethylation in regulating B56 holoenzyme formation, we specifically immunoprecipitated B56α (Fig. 1b) and B55α/δ (Fig. 1c) holoenzymes from wild-type HAP1 and HAP1 *Lcmt1* null cells. In PP2A-C methylation proficient wild-type HAP1 cells, both regulatory subunits exclusively coimmunoprecipitated methylated PP2A-C over non-methylated PP2A-C[35]. In *Lcmt1* null cells, in which only non-methylated PP2A-C is present, trimeric holoenzymes with B55α/δ as well as with B56α were substantially reduced down to ~1/3 of the levels in wild-type cells. In case of the B55 subunit the holoenzyme reduction was primarily caused by a 60% reduction in the total cellular B55α/δ levels confirming previous findings[21,36], and only to a lesser extent by a reduced ability to form a complex with PP2A-C and the structural A subunit (PP2A-A). This was different for the B56α subunit, which showed similar total expression levels in wild-type and *Lcmt1* null cells. However, B56α had a reduced ability to interact with non-methylated PP2A-C/PP2A-A dimers indicating a high methyl-PP2A-C affinity/preference of B56α. Re-expression of V5-tagged LCMT1 in *Lcmt1* null cells resulted in a quantitative rescue of the defects in PP2A-C methylation as well as in B56 holoenzyme assembly (Supplementary Fig. 1a), B55 expression and B55 holoenzyme formation (Supplementary Fig. 1b). The reduction in the abundance of methylation-sensitive trimers in the *Lcmt1* null cells correlates with increased phosphorylation of Ser9 within GSK3beta, a downstream substrate of the PI3K-AKT pathway. Conversely, the increase in the abundance of methylation-sensitive trimers upon LCMT1 re-expression correlated with the dephosphorylation of GSK3beta (Supplementary Fig. 1c) confirming the negative regulation of the PI3K-AKT pathway by methylation-sensitive PP2A holoenzymes. Through their ability to negatively regulate the PI3K-AKT, p70/p85 S6K, ERK/MAP kinase, and the c-MYC pathways, methylation-sensitive PP2A enzymes are thought to act as tumor-suppressors, whose downregulation/mutation promotes tumorigenesis[17]. We therefore hypothesized that changes in the levels of PP2A-C α−carboxymethylation could result in biased heterotrimer formation thus serving as a marker for the levels of tumor suppressive

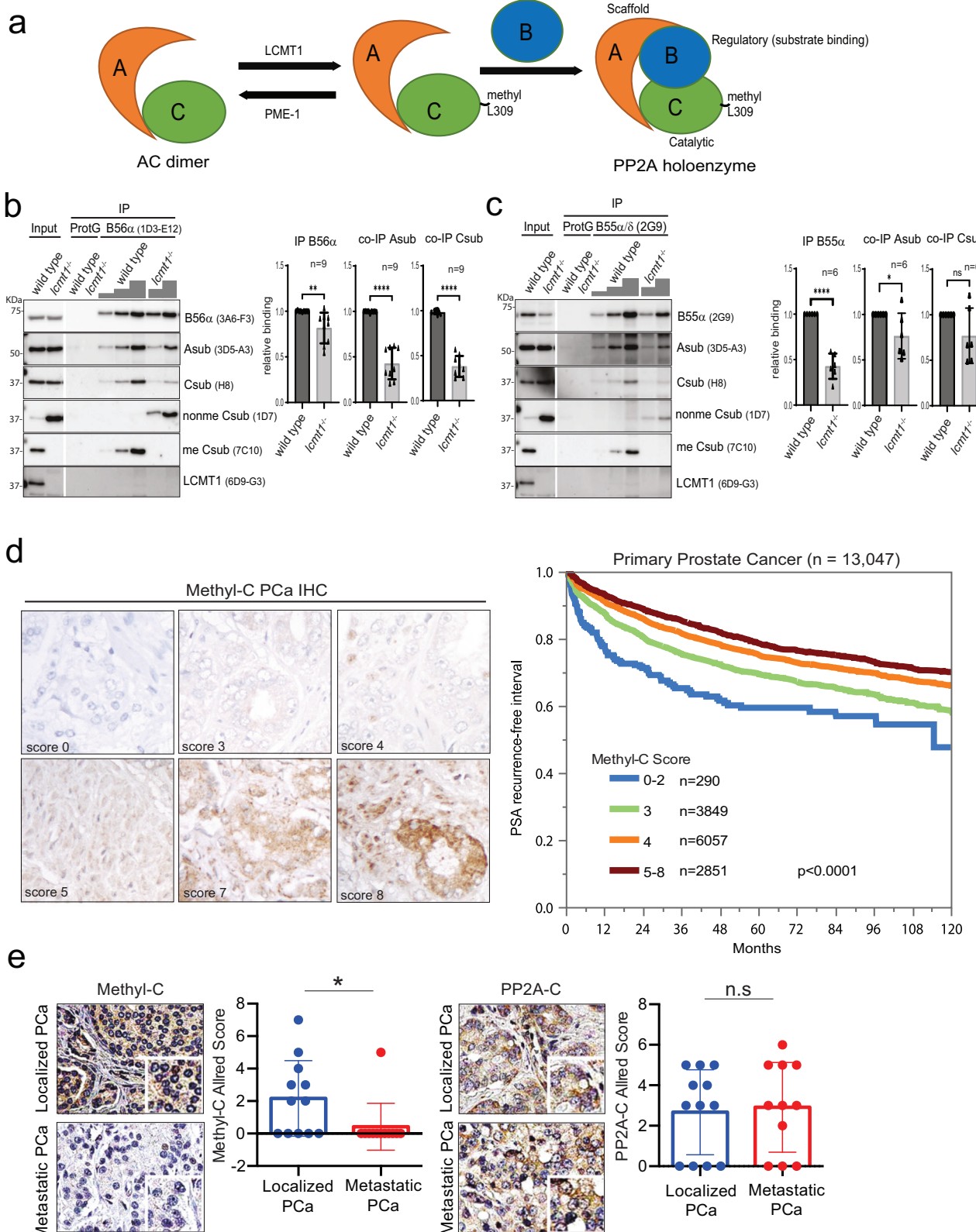

PP2A activity in cells and tumor tissues and as both a predictive and/or prognostic biomarker for patient outcome and treatment response. To test this hypothesis, we chose as a model system prostate cancer because hyperactivation of the PI3K-AKT-mTOR pathway has been associated with disease progression from the primary to metastatic phase and carboxymethyl sensitive PP2A regulatory subunits regulates this pathway[24,37,38].

We previously generated a mouse monoclonal antibody 7C10-C5, whose further characterization revealed high specificity for methylated PP2A-C and no cross-reactivity with the related phosphatases PP4 and PP6 that share the YFL motif at their carboxy-termini[35] (Supplementary Fig. 1d–g). 7C10-C5 stained prostate cancer tissues in a methyl-PP2A-C specific manner as the immunohistochemical signal could be blocked by prior incubation of 7C10-C5 with a methylated but

**Fig. 1 | Methyl-PP2A-C loss is associated with prostate cancer progression.**
**a** Schematic showing methylation-sensitive PP2A holoenzyme complex.
**b, c** Reduced levels of B56 and B55 subunit-containing holoenzymes in cells lacking PP2A methylation as a result of CRISPR mediated knockout of LCMT1. Immuno-blotting of lysates and anti-B56α or anti-B55α/δ immunoprecipitates from lysates of HAP1 wildtype and lcmt1−/− cells. 1/10 of the input lysate was loaded for the B56α blot, and 1/160 of the input lysate for all other blots. The panels originate from 3 independent blotting membranes, which were sequentially incubated with the indicated antibodies. The blots are representative of n = 9 (B56α) and n = 6 (B55α/δ) independent immunoprecipitation experiments. The amounts of immunoprecipitated B56α and B55α/δ were quantified, and the coimmunoprecipitated A and C subunit levels were normalized to the B56α or B55α/δ levels, which were set to 1 for the wild-type cells. Statistical significance of quantified protein levels was assessed using an unpaired two-sided Student's t-test. **b** **P = 0.0048 ****P < 0.0001 **c** ****P < 0.0001 *P = 0.0434, ns P = 0.0932. Data are presented as mean values ± s.d. **d** Representative images of methyl-C immunohistochemistry in primary prostate cancers. From upper left to lower right: score 0, 3, 4, 5, 7, and 8. A low immuno-histochemistry methyl-C score is linked to early biochemical recurrence in the tissue microarray cohort of 13,047 prostate cancers with available follow-up data. **e** Representative immunohistochemistry images showing expression of methyl-PP2A-C (methyl-C) and total PP2A-C in localized and metastatic prostate cancer tissues. Bar graphs show the quantification, error bars mean ± s.d., n = 11 metastatic tissue and n = 12 localized tissue samples. Student two-tailed t-test, p value: 0.0376 (methyl-C), p value: 0.7902 (PP2A-C). *P < 0.05, **P < 0.01, ***P < 0.001 ***P < 0.0001, ns non-significant.

not with a non-methylated PP2A-C C-terminal peptide (Supplementary Fig. 2a). In addition, the IHC specific signal was abolished by the chemical removal of the PP2A-C methylester with NaOH treatment of the tissue sample. Next, we conducted IHC with the 7C10-C5 in a series of tissue microarrays (TMAs) representing 17,747 localized PCa across all Gleason scores (Supplementary Table 1). In this cohort 13,047 prostate cancer patients had available follow-up data, and we observed a significant correlation between low methyl-PP2A-C scores and biochemical recurrence (Fig. 1d). Additionally, low methyl-PP2A-C levels were negatively associated with critical parameters of disease progression including high Gleason score, positive surgical margins, and lymph node metastasis (Supplementary Table 2). These differences were not correlated with TMPRSS2-ERG rearrangement (Supplementary Fig. 2b, c). IHC validation was then performed in an independent cohort of Gleason score 6 and 7 prostate cancer tissues and again there was a significant association between early disease recurrence and low methyl-PP2A-C scores in the Gleason score 7 group indicating that at this disease stage the loss of PP2A-C methylation might indeed be linked to the progression to metastatic disease and treatment resistance (Supplementary Fig. 2d–f). Importantly, 7C10-C5 staining of a localized and metastatic prostate cancer TMA revealed a complete loss of PP2A-C carboxymethylation specifically in metastatic prostate cancer cells and as expected a more heterogenous pattern of staining noted in localized prostate cancer specimens (Fig. 1e, Supplementary Fig. 2g). Interestingly, a recent proteome analysis of localized and metastatic prostate cancer found a strong negative correlation between LCMT1 levels and AR in metastatic prostate cancer (Supplementary Fig. 2h, i)[39].

LCMT1 loss activates AR signaling and promotes castration-independent tumor growth. Hyperactive AR, in association with critical coactivators such as p300, BRD4, and MED1, drives prostate cancer progression[29,40,41]. Increased phosphorylation of AR and associated coactivators are critical for stable chromatin interaction and transcription as the disease progress[29,42]. To investigate the phosphorylation status of AR and MED1 as a consequence of methyl-PP2A-C loss and the resultant bias in PP2A heterotrimer formation upon LCMT1 silencing, we employed a panel of prostate cancer cell lines stably expressing shRNA to LCMT1. The knockdown of LCMT1 led to reduced total PP2A-C as well as methyl-PP2A-C expression in the nucleus and cytoplasm accompanied by high levels of phosphorylated AR and MED1, as well as MYC, a known PP2A substrate (Fig. 2a, Supplementary Fig. 3a). Next, CoIP of PP2A-A in LCMT1 silenced cells resulted in a reduced abundance of methylation-sensitive AB56αC and AB55αC heterotrimers, whereas PP2A complex formation involving methylation insensitive PR130 and STRN regulatory subunits was unaffected (Fig. 2b, Supplementary Fig. 3b). This finding confirms the observed result in HAP1 cells (Fig. 1b, c) and suggests that targeted reduction of carboxymethylation through LCMT1 depletion results in altered heterotrimer assembly, specifically of B56α and B55α subunit binding to the AC dimer. Recent reports suggest that PP2A affects many key aspects of RNA PolII transcription, including pause

regulation, elongation, and termination[43], however it remains unclear whether specific PP2A heterotrimers regulates transcription by directly targeting chromatin-associated factors for dephosphorylation. Interestingly, chromatin fractionation in LNCaP cells demonstrated the presence of canonical AB56αC heterotrimer on chromatin, which was reduced upon LCMT1 silencing (Fig. 2c). Next, we hypothesized that the reduction in PP2A chromatin-bound heterotrimers in LCMT1 silenced cells could result in the accumulation of phospho-substrates even in the absence of ligand-dependent activation. To test this hypothesis, cells were starved of androgen for six days, followed by chromatin extraction. This resulted in a marked loss of p-AR/AR and p-MED1/MED1 on the chromatin. Interestingly, under this context of androgen deprivation there was an increased association of the AB56αC heterotrimer with the chromatin which was attenuated in the LCMT1 silenced cells (Fig. 2d). This was further corroborated by the continued transcription of canonical AR target genes (Fig. 2e, Supplementary Fig. 3c), and increased growth, and proliferation of LCMT1 silenced cells under conditions of androgen deprivation (Fig. 2f). Moreover, aggressive growth and resistance to the antiandrogen enzalutamide was observed upon LCMT1 silencing; however, these cells were still sensitive to AR degradation by AR-specific PROTAC, suggesting a continued AR dependency (Supplementary Fig. 3d–g). To further demonstrate that LCMT1 loss promotes castration-independent growth in vivo, we performed a xenograft experiment with LNCaP cells stably expressing shLCMT1 or shNTC in naïve and castrated mice. Interestingly, there were similar tumor take rates (approximately 90%) in naïve non-castrated mice with both cell lines. Strikingly, only LCMT1 silenced cells developed tumors in castrated mice with a tumor uptake of 83% compared to a mere 20% with the control cells (Fig. 2g). Most importantly, the tumors derived from shLCMT1 cells in castrated animals displayed continued AR activity without any evidence of neuroendocrine differentiation (Supplementary Fig. 3h). Together, these data demonstrate that the loss of methylation-sensitive PP2A heterotrimers from the chromatin upon LCMT1 silencing results in hyperphosphorylation mediated amplified AR-signaling leading to ligand-independent AR addiction, a hallmark of prostate cancer.

AB56αC stabilization by LCMT1 targets MED1-AR for dephosphorylation. To investigate whether LCMT1 mediated PP2A heterotrimerization directly affects chromatin-specific p-MED1 and p-AR, we generated LCMT1 null 293 T cells (polyclonal pool) using CRISPR. Ectopic expression of HA-tagged MED1 and Halo-tagged AR showed increased p-MED1 and p-AR in the LCMT1 null cells compared to Cas9 control, and interestingly, the phosphorylation of endogenous MED1 was substantially higher in the null cells, suggesting a direct role of LCMT1 via PP2A in regulating MED1 and AR phosphorylation (Fig. 3a). Next, in an orthogonal approach, LCMT1 overexpression led to an increase in methyl-PP2A-C, resulting in a more stable AB56αC trimer with a concomitant reduction in p-MED1 and p-AR (Fig. 3b, Supplementary Fig. 4a). Since a large majority of MED1 and AR exists in transcriptionally active superenhancers[29,44]—we reasoned that

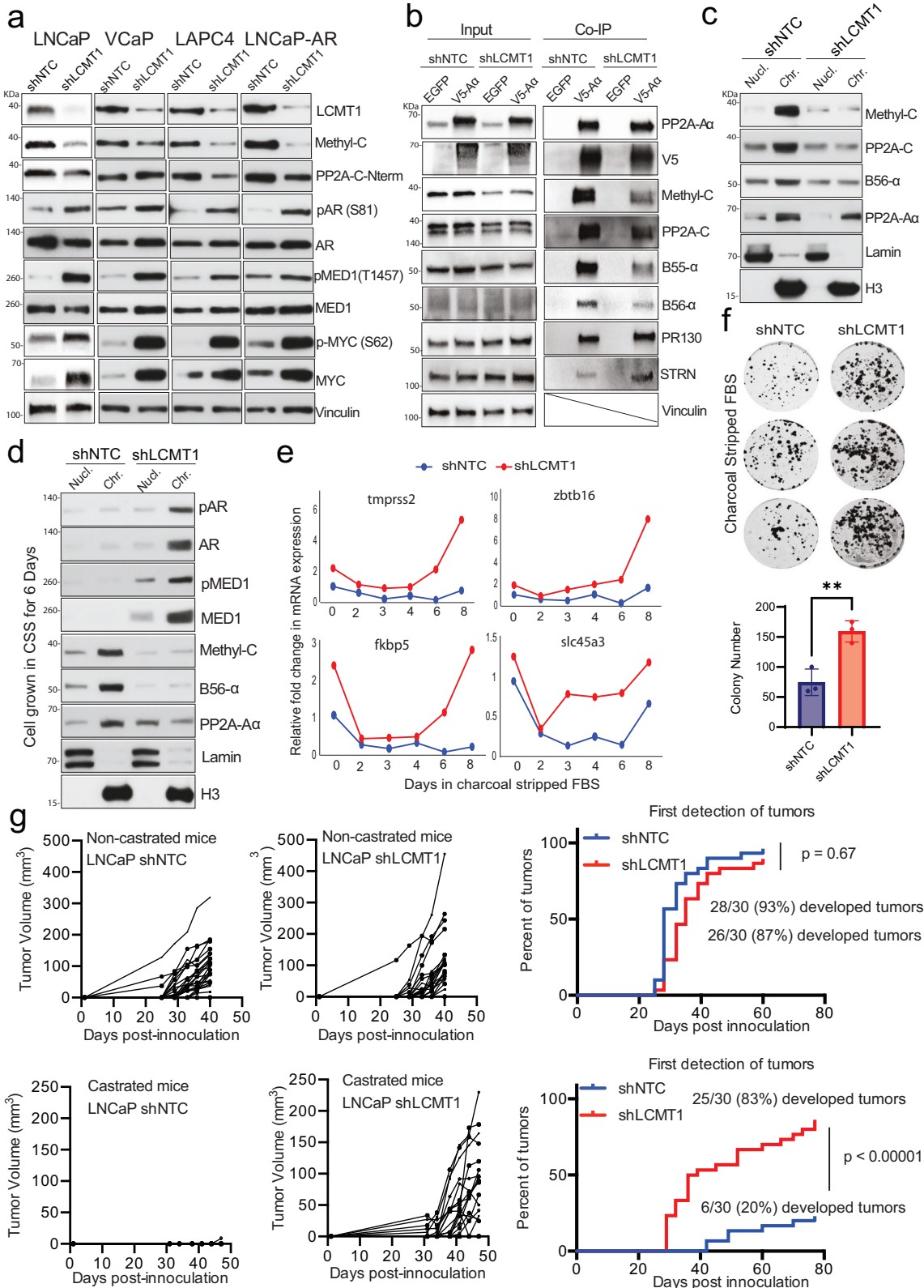

chromatin-specific p-MED1 and p-AR could be visualized as discrete puncta in the nuclei of cells. Fixed cell immunofluorescence (IF) with antibodies against p-MED1 and p-AR revealed nuclear puncta for both factors in LNCaP cells expressing LacZ, however, a significant reduction in the puncta and total fluorescence accompanied by increased methyl-PP2A-C staining was observed in cells overexpressing LCMT1 (Fig. 3c). Next, chromatin fractionation in the LCMT1 overexpressing

cells confirmed the accumulation of AB56αC heterotrimer with a parallel reduction in p-MED1 and p-AR, suggesting LCMT1 not only stabilizes specific PP2A holoenzymes but also enhances their ability to bind chromatin in a methyl-PP2A-C dependent manner resulting in AR and MED1 dephosphorylation (Fig. 3d). Since PP2A engages its substrates through a specific regulatory B subunit, we next asked whether AR and MED1 could be the direct target of B56α. The in silico sequence

**Fig. 2 | LCMT1 loss activates AR signaling and promotes castration-independent tumor growth. a** Methyl-C reduction upon LCMT1 silencing results in a concomitant increase in p-AR, p-MED1, and MYC expression. Representative immunoblots in a panel of prostate cancer cells stably expressing shNTC or shLCMT1 showing indicated proteins. Vinculin was used as a loading control. **b** Reduced AB56αCme and AB55αCme (PP2A) heterotrimers in LCMT1 silenced LNCaP cells. Coimmunoprecipitation (IP) analysis using lysates from the indicated cells with V5-PP2A-Aα and EGFP-specific antibodies, demonstrating the interaction between PP2A-Aα, methyl-PP2A-C, B56-α, B55-α. PR130 and STRN are used as methylation insensitive controls and EGFP as a negative IP control. Total lysate was used as input control. **c** LCMT1 loss disrupts canonical AB56αCme heterotrimer association with chromatin. Chromatin and soluble nuclear fractions from control and LCMT1 silenced LNCaP cells were subjected to immunoblotting for the indicated proteins. Lamin and H3 served as controls for the nuclear and chromatin fractions, respectively. **d** LCMT1 abrogation resists androgen-ablation arbitrated suppression of AR and MED1 recruitment to the chromatin. Chromatin and soluble nuclear fractions extracted from the cells grown in charcoal-stripped fetal bovine serum (CSS) medium for six days were subject to immunoblotting for the indicated proteins. Lamin and H3 served as controls for nuclear and chromatin fractions, respectively. **e** Continuous AR transcriptional activity under androgen deprivation. qRT-PCR for AR-regulated genes in cells grown in CSS-containing medium. **f** Colony formation assay demonstrates the increased proliferation of LCMT1 depleted cells in hormone-deprived media. Cells were cultured in CSS media for 10 days, followed by crystal violet staining (top), and quantification (bottom), error bars mean ± s.d., $n = 3$ biologically independent replicates. Students two-tailed $t$-test, $p$ value: 0.006. **g** Castration-independent tumor growth in vivo upon LCMT1 silencing. Castrated and non-castrated mice bearing shLCMT1- or shNTC-LNCaP were evaluated for tumor growth. The tumor volume is plotted (left) and the percentage of tumors first detected is shown (right). Fisher's exact test of relative risk. The data in **a**–**d** are representative of $n = 3$ independent experiments.

analysis of all 30 subunits of the human Mediator complex showed the presence of two LxxIxE small linear motifs (SLiMs) on MED1—a newly identified conserved surface-exposed docking motif for B56α[45], however, no such SLiM was found in AR (Fig. 3e). The two functionally characterized and critical threonine phosphorylation sites (T1032 and T1457) in MED1 are present in its large intrinsically disordered region (IDR)[29]. Interestingly, the two B56α binding SLiMs were also localized to MED1 IDR at 525-533 "a" and 783-791 "b", which both showed a near perfect consensus sequence to other validated PP2A-B56α substrates such as BUB1 and CDCA2, and conservation across species (Supplementary Fig. 4b, c). Of note, the site "b" has acidic residues inside and immediately C-terminal to the motif, a characteristic associated with higher affinity B56α binding[45]. To test the potential for direct interaction between MED1 and B56α, we mutated the two degenerate yet highly conserved amino acid residues, critical for binding to B56α, in the SLiM sequences found in MED1 at positions one (leucine) and four (isoleucine) to alanine (2A) (Fig. 3f). Co-transfection of B56α with either wild-type HA-MED1 (wt) or HA-MED1-SLiM mutants followed by B56α CoIP showed co-purification of wt-MED1 and AC dimer, whereas the 2A mutation at either Site a (MED1-2A-a) or Site b (MED1-2A-b) reduced binding and this interaction was further abolished when both the sites were mutated (MED1-2A-ab) (Fig. 3g). These observations were further confirmed with purified GST-B56α protein that showed interaction with endogenous MED1 (Supplementary Fig. 4d), which exclusively pulled-down wt-MED1 but not the MED1-2A-ab mutant (Fig. 3h). Collectively, this data suggests a direct interaction of MED1 with B56α through the conserved LxxIxE SLiMs, and we therefore next reasoned that compared to MED1-SLiM mutants, wild-type MED1 would be uniquely sensitive to dephosphorylation by LCMT1 stabilized AB56αC heterotrimers. To investigate this, we performed CoIP with cells cotransfected with LCMT1 and HA-tagged wild-type or SLiM mutant MED1 constructs and probed with antibodies against p-MED1. Overexpression of LCMT1 led to a decrease in phosphorylated wild-type MED1; however, MED1-SLiM site "b" and double mutant displayed phosphorylated signal similar to vector controls (Fig. 3i), thereby validating MED1 as a bona fide B56α substrate and suggesting that disruption of this interaction could lead to accumulation of phosphorylated MED1 in complex with AR on chromatin. Next, we sought to define the mechanistic basis for the observed LCMT1-PP2A-B56α mediated changes in p-AR levels though no LxxIxE SLiMs were identified on the androgen receptor (Figs. 2a, d and 3a–e). We have recently reported that the interaction between MED1 and AR is phosphorylation-dependent, and that inhibition of CDK7-mediated phosphorylation of MED1 results in derecruitment of the chromatin-bound MED1-AR complex and loss of AR-mediated transcription[29]. Therefore, to test whether AB56αC heterotrimer mediated AR dephosphorylation could occur in a MED1-dependent manner, we cotransfected LCMT1 and Halo-tagged AR with HA-tagged wild-type or SLiM mutant MED1 in HEK293-T cells followed by HA CoIP. As expected, the MED1-SLiM mutant displayed increased phosphorylation. Compared to wild-type MED1, a noticeably higher degree of interaction and phosphorylation of AR was observed with the MED1-SLiM mutant (Fig. 3j), thus suggesting that the stabilization of methylation-sensitive PP2A-B56α heterotrimers by LCMT1 results in MED1-dependent dephosphorylation of AR (Fig. 3k).

LCMT1 inhibits AR-MED1 transcriptional activity and prostate cancer growth. Following the observation that stabilization of PP2A-B56α holoenzyme by LCMT1 targets AR and MED1 for dephosphorylation, we sought to study the effect of LCMT1 on the AR and MED1 cistrome and subsequent transcriptional output of the AR-MED1 complex. Similar to the observed loss of chromatin-associated p-AR/AR and p-MED1/MED1 (Fig. 2d), chromatin immunoprecipitation followed by sequencing (ChIP-seq) revealed a genome-wide decrease of AR and MED1 binding in LNCaP cells overexpressing LCMT1 (Fig. 4a, Supplementary Fig. 5a). To examine the nature of AR- and MED1-bound sites, we first identified transcriptionally active regions using H3K27ac ChIP-seq in LNCaP cells (GSM1902615) and nominated enhancers and superenhancers (Supplementary Fig. 5b). Interestingly, significantly reduced AR and MED1 enrichment was found in the enhancers and superenhancer regions upon LCMT1 overexpression (Supplementary Fig. 5c). The loss of AR and MED1 signals was evident from the H3K27ac and BRD4 bound regulatory sites of the canonical AR target gene *klk3* and others (Fig. 4b, Supplementary Fig. 5d). These observations together demonstrate that the LCMT1 mediated formation of PP2A-B56α heterotrimers targets AR and MED1 for dephosphorylation, resulting in their derecruitment from the enhancers and superenhancers. Consequently, global transcriptomic profiling with RNA sequencing (RNA-seq) showed differential expression of hundreds of genes in these cells (Fig. 4c). Next, gene set enrichment analysis (GSEA) further revealed significant downregulation of AR-regulated genes (Fig. 4c, d, Supplementary Fig. 5e). Interestingly, negative enrichment of MYC-regulated genes without a change in its transcript expression also was observed (Supplementary Fig. 5e), which further confirms MYC protein regulation by the PP2A-B56α heterotrimer[46] (Figs. 4c and 3b). Furthermore, LCMT1 cells displayed a negative enrichment of the hallmark PI3K-AKT-mTOR signaling genes, which is in accordance with previous observations of effectors of the PI3K-AKT signaling pathways as being targets of methyl-sensitive PP2A heterotrimers (Fig. 4d)[20]. This negative impact on the transcriptional pathways regulated by the three most critical drivers of prostate cancer -AR, MYC, and AKT[24], was also reflected in the phenotypic response of the LCMT1 expressing cells in both proliferation and long-term colony formation assays as well as in their increased sensitivity to enzalutamide treatment (Supplementary Fig. 5f–g). Next, to determine the effects of LCMT1 on tumor growth in vivo, we carried out a xenograft experiment in mice, and observed significantly reduced tumor growth in the LCMT1 overexpressing cells compared to their

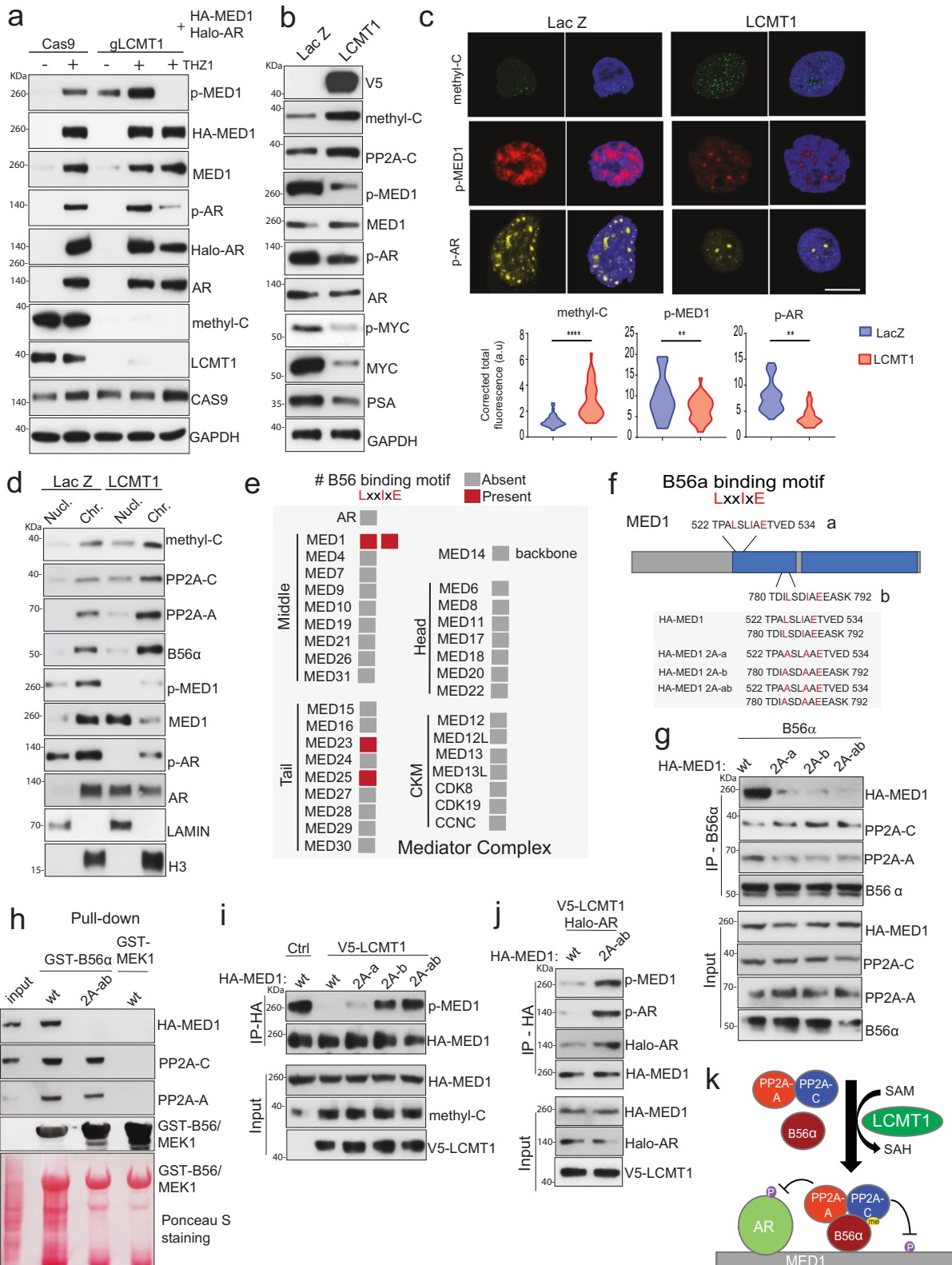

respective control (Fig. 4e). Specifically, there was an 87% (13/15) tumor uptake in the LNCaP control cells, compared to only a 7% (1/15) tumor take rate in LCMT1 overexpressing LNCaP cells. We were able to validate these observations in an independent cell line based model in which, control VCaP cells had an 80% (16/20) tumor uptake in vivo, as compared to a 10% (2/20) tumor take rate in LCMT1 overexpressing VCaP cells in non-castrated mice (Fig. 4e). Together,

these findings provide evidence that LCMT1 is both a critical regulator of AR-MED1 transcriptional activity and a potent suppressor of oncogenic pathways driving prostate tumor growth in vivo.

**S6K1- and β-TrCP mediated degradation of LCMT1.** Although LCMT1 is the only enzyme known to methylate the C subunit of PP2A AC dimer and the catalytic subunits of the PP4 and PP6 family protein phosphatase[21,22], not much is known about the regulation of this critical

**Fig. 3 | LCMT1 abrogates AR-MED1 transcriptional activity by stabilizing the AB56αCme heterotrimer on chromatin. a** LCMT1 loss leads to the accumulation of hyperphosphorylated AR and MED1. Cas9 control and LCMT1 null HEK293-T cells were co-transfected with HA-MED1 and Halo-AR for 48 h or treated with 100 nM THZ1 (CDK7i) for 12 h followed by immunoblotting for the indicated proteins. **b** LCMT1 overexpression increases methyl-C levels with a concomitant reduction in AR-MED1 phosphorylation. Protein lysates were subjected to immunoblotting for the indicated proteins. **c** Representative confocal images show nuclear methyl-C expression is inversely correlated with p-MED1 and p-AR expression (top). Quantification (bottom) is in the form of corrected total cell fluorescence, with two-tailed *t*-tests *p < 0.05, **p < 0.01. Scale bar: 10 µm. **d** Immunoblotting showing enhanced recruitment of AB56αCme and loss of AR and MED1 on the chromatin upon LCMT1 overexpressing. Lamin and H3 served as controls. **e** Color-coded plot showing the presence or absence of B56α binding SLiMs on AR and subunits of human Mediator complex. **f** Schematic depicting two B56α binding motifs on MED1. The blue region

indicates IDR – intrinsically disordered region. MED1 mutants used in the study are shown. **g** MED1 binds PP2A-B56α through two conserved LxxIxE motifs. B56α (B56a) was co-transfected with either HA-MED1-wild type (wt) or the indicated 2A mutant constructs in HEK293-T cells for 72 h post transfection, cell lysates were subjected to co-IP with the B56α antibody for the shown proteins. **h** GST-pull-down assay showing purified GST-B56α binds to MED1-wt not 2A-ab mutant. GST-MEK1 was used as a negative control. **i** HEK293-T cells were transfected with either MED1-wt or the indicated MED1 mutant alone or with pLX304-V5-LCMT1 for 72 h, followed by co-IP and immunoblotting for the indicated proteins. **j** Dephosphorylation of AR by PP2A-B56α is through MED1 binding. Co-IP with HA antibody was performed using lysates prepared from HEK293-T cells transfected with Halo-AR and V5-LCMT1 constructs and either HA-MED1-wt or HA-MED1-2A-ab. **k** Schematic showing LCMT1 mediated PP2A-B56α heterotrimerization directly targets MED1-AR phosphorylation. The data in **a**, **b**, **d**, **g–j** are representative of at least *n* = 3 independent biological replicates.

phosphatase regulating enzyme. Analysis of the TCGA prostate dataset revealed no significant change in the expression of LCMT1 mRNA or any mutation in its coding sequence or genomic deletion in primary prostate cancer and benign prostate tissue (Supplementary Fig. 6a). Additionally, pan-cancer analysis in TCGA revealed a comparable level of RNA expression for LCMT1 between normal and cancer tissues across all cancer types (Supplementary Fig. 6b), suggesting a potential post-translational mechanism of regulation for this enzyme (Supplementary Fig. 6c). Indeed, a negative correlation between LCMT1 protein levels and AR has been identified in metastatic prostate cancer[39]. Treatment of LNCaP cells with proteasome inhibitor (MG132) or cullin neddylation inhibitor (MLN4924) led to the accumulation of LCMT1, implying a potential cullin-RING E3 ubiquitin ligases (CRLs) dependent mechanism in regulating LCMT1 stability and degradation (Fig. 5a). Substrate phosphorylation has been shown to regulate interaction with CRLs leading to ubiquitination and degradation by the proteasome machinery[47]. To address a potential phospho-regulatory mechanism of LCMT1, we first tested if it exists in a phosphorylated form. Using the Phos-tag immunoblot assay, in which the dinuclear metal complex 1,3-bis[bis(pyridin-2-ylmethyl)amino]propan-2-olato dizinc(II) specifically binds to phosphorylated amino acids and generates a mobility shift on acrylamide gels proportional to incorporated phosphates[29,48], we observed a slow-migrating band(s) for LCMT1, which was eliminated upon phosphatase (Calf intestinal alkaline phosphatase (CIAP)) treatment (Fig. 5b). This was further supported by publicly available high throughput mass-spectrometry (HT-MS) data sets, which displayed Serine (S)-9 as the most frequently phosphorylated residue, followed by S12 and S8 in LCMT1 (Supplementary Fig. 6d). This observation prompted us to examine the LCMT1 AlphaFold-predicted structure[49], which showed its N-terminal region containing these serine residues to be an IDR -known to frequently undergo post-translational modifications (PTMS) such as phosphorylation[50]. Noticeably, we identified a conserved phosphorylation consensus R/X/R/X/X/S/T sequence (R[4]QRESS[9]), for the protein kinase AKT-mTOR-ribosomal protein S6 kinase 1 (S6K1) in its N-terminal IDR (Fig. 5c). Intriguingly, the residues shown to be phosphorylated in the HT-MS data were either present in the canonical AKT/S6K1 site (S8/S9) or immediately following it (S12). To gain insight into the pathways involved in the phosphorylation-mediated degradation of LCMT1, we treated LNCaP cells with various PI3K-AKT-mTOR inhibitors. Accumulation of LCMT1 and a parallel increase in methyl-PP2A-C was evident in all the drug treated cells (Fig. 5d). The observation that Torin and Rapamycin, an inhibitor of mTOR, but not phosphoinositide 3-kinase (PI3K) or the protein kinase AKT (which are both upstream to mTOR), had the same effect as LY294002−an inhibitor of PI3K, indicated that the mTOR-ribosomal protein S6 kinase 1 (S6K1) could be the potential kinase involved in promoting LCMT1 degradation. Since growth factors and cytokines activate the AKT-mTOR pathway, we examined the impact of serum starvation followed

by EGF stimulation on LCMT1 expression. LNCaP cells and HEK293-T cells were serum starved for 24 h and then stimulated by the addition of EGF for varying time points. In the serum-starved cells, LCMT1 level was higher than in cells grown in complete medium, but after EGF stimulation, it decreased rapidly, paralleling AKT-mTOR signaling activation and methyl-PP2A-C loss (Fig. 5e). Additionally, treatment of these cells with AKT-PI3K inhibitors but not the MEK inhibitor Trametinib completely reversed the effects of EGF, ruling out any role of the MAPK pathway in regulating LCMT1 protein turnover (Supplementary Fig. 6e). Involvement of mitogen-activated PI3K-AKT-mTOR in the degradation of LCMT1 was further supported by the observed loss of LCMT1 and methyl-PP2A-C in cells expressing constitutively active AKT (Supplementary Fig. 6f). Next, we explored the potential S6K1 phosphorylation sites on the LCMT1 in the context of mitogen-induced LCMT1 degradation. We generated a number of LCMT1 mutants (all with V5-tags) in which Ser8, Ser9, or Ser12 were mutated individually or together to Ala (e.g., Ser8 to Ala, S8A). Though the wild-type LCMT1 and the LCMT1 (S8A), LCMT1 (S9A), LCMT1 (S12A), and LCMT1 (S8/9/12A) mutants remained stable in HEK293-T cells under serum-starved condition, EGF stimulation led to degradation of the wild type but not the mutant LCMT1 proteins (Fig. 5f). These findings indicate that the serine residues in the N-terminal IDR of LCMT1, through a conserved phosphorylation-dependent mechanism involving S6K1, play a crucial role in the regulation of its protein stability. To determine whether these serine residues in LCMT1 indeed undergo phosphorylation, after expressing wild-type and mutant proteins in HEK293-T cells, we immunoprecipitated them with V5 antibody and probed with a phosphospecific antibody that recognizes the S6K1 phospho-consensus motif RXRXXpS/T[51]. Although this antibody recognized the wild-type LCMT1, a reduced signal was observed for single mutants, and the LCMT1-S8/9/12 A mutant displayed the most severe loss of signal (Fig. 5g). Furthermore, treatment of cells with Torin or siRNA-mediated knockdown of S6K1 reduced phosphorylated LCMT1 (Fig. 5h) and led to its stabilization under mitogen stimulated condition (Fig. 5i), further supporting a role for S6K1 in regulating LCMT1 expression and degradation. Importantly, the interaction between LCMT1 and S6K1 was confirmed by reciprocal immunoprecipitation using FLAG-S6K1 and V5-LCMT1 (Supplementary Fig. 6g), suggesting LCMT1 as a direct substrate for S6K1.

Having established that LCMT1 is phosphorylated by S6K1, and that this phosphorylation event promotes LCMT1 degradation, we sought to identify the relevant E3-ligase involved in such mechanism. To this end, we noticed that LCMT1 phosphorylation sites follow the canonical pattern of β-TrCP targets (Supplementary Fig. 6h). In a typical β-TrCP substrate, the two serine residues in the DSGXXS consensus motif must be phosphorylated to allow recognition[51]. The putative β-TrCP binding motif in LCMT1, with Ser9 and Ser12, is immediately preceded by Ser8, which upon phosphorylation, could potentially substitute for the aspartic acid residue of the canonical

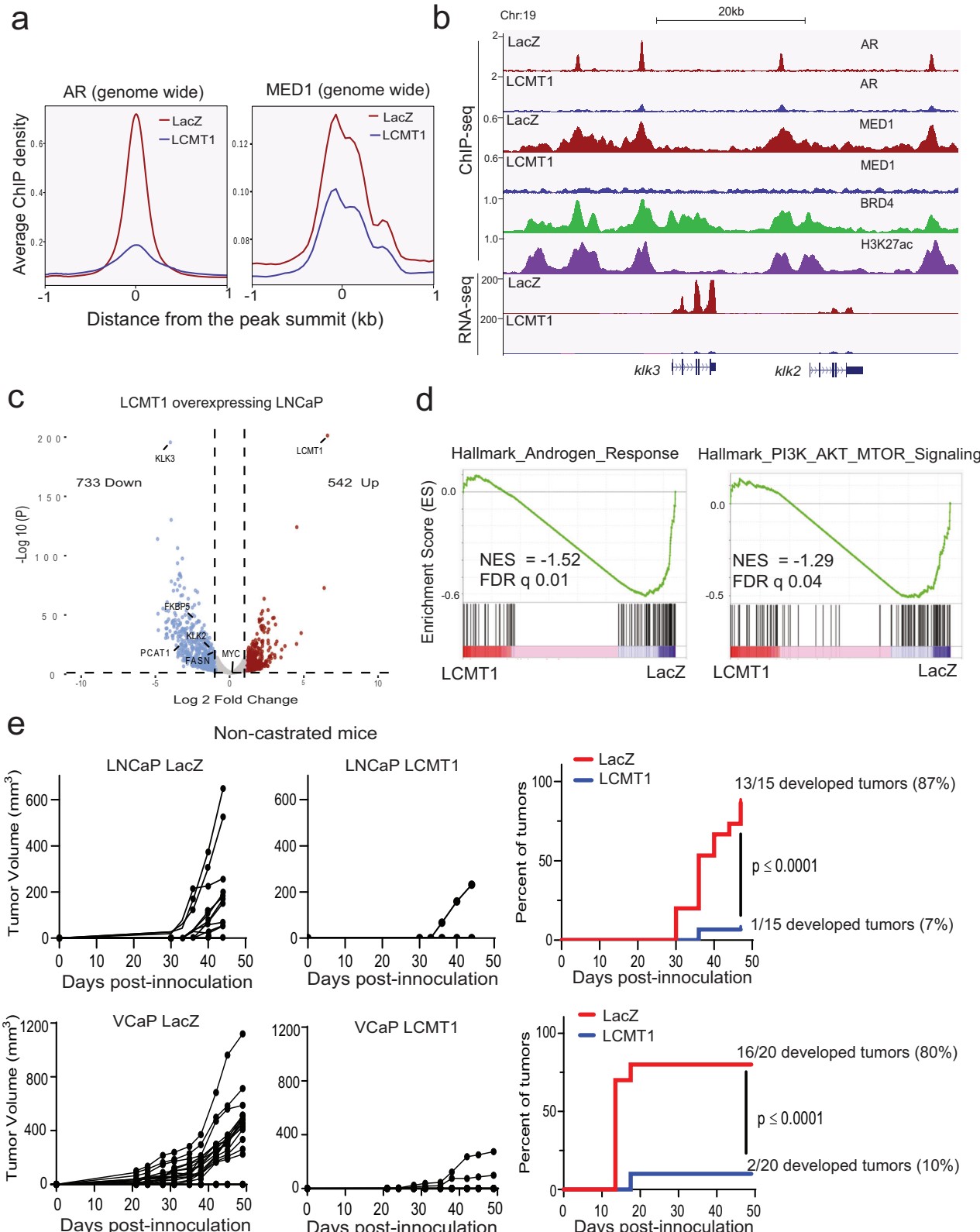

**Fig. 4 | LCMT1 abrogates AR-MED1 transcriptional activity and prostate cancer growth. a** LCMT1 over-expression decreases AR and MED1 recruitment to the chromatin. Genome-wide averaged AR and MED1 ChIP-seq enrichment profile plot in LNCaP cells stably overexpressing LCMT1 or LacZ. **b** Genome browser tracks of AR and MED1 binding at *klk2* and *klk3* loci in the indicated cells, and BRD4/H3K27ac tracks from LNCaP cells. The tracks at the bottom show the RNA-seq gene expression. **c** Volcano plot of RNA-seq results showing differentially expressed genes in LCMT1 versus LacZ expressing LNCaP cells. **d** GSEA plots showing negative enrichment of Hallmark Androgen Response and PI3K/AKT/mTOR pathway genes in LCMT1 overexpressing LNCaP cells. NES normalized enrichment score, FDR false discovery rate. **e** LCMT1 suppresses in vivo tumor growth. Naïve mice bearing either LNCaP or VCaP-LCMT1 and -LacZ xenografts were evaluated for tumor growth for the given number of days. The tumor volume (measured twice per week using calipers) and the percentage of tumors detected is shown. *P* value indicates Fisher's exact test of relative risk.

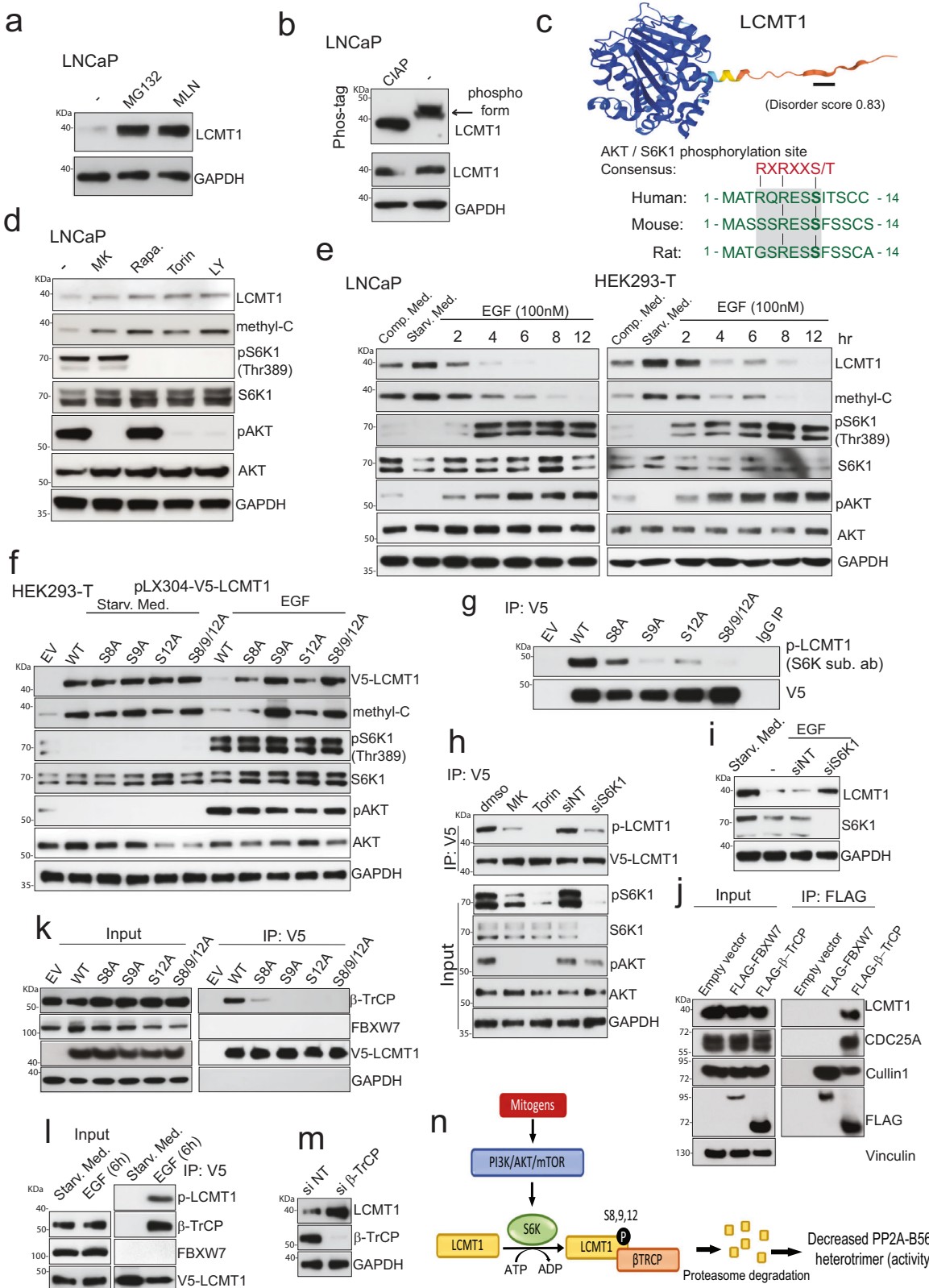

degron. To test for β-TrCP binding to LCMT1, we performed a FLAG-immunoprecipitation assay from cell lysates expressing either FLAG-TrCP1 or FLAG-FBXW7 as negative control (Fig. 5j). Strikingly, endogenous LCMT1 specifically interacted with β-TrCP but not FBXW7. Cdc25 served as a positive control for β-TrCP;[52] CULLIN1 is a common partner of both FBXW7 and β-TrCP. To test whether LCMT1 binds β-TrCP via the pSpSXXpS motif, we tested the binding of LCMT1 Ser8/

9/12A mutants to endogenous β-TrCP1. Single mutations of S8, S9, and S12 to A or the triple S8/9/12/A mutation abrogated the interaction between LCMT1 and endogenous β-TrCP (Fig. 5k). CoIP in cells transfected with V5-LCMT1 under EGF stimulation confirmed the interaction between pLCMT1 and endogenous β-TrCP (Fig. 5l). Importantly, the expression of siRNAs targeting both BTRCP1 and BTRCP2 revealed upregulation of the endogenous LCMT1 protein (Fig. 5m), and

**Fig. 5 | Phosphorylation-dependent degradation of LCMT1 mediated by S6K1- and β-TrCP. a** Immunoblotting showing MG132 and MLN4924 (10 μM, 12 h) restore LCMT1 protein expression. **b** Phos-tag analysis showing LCMT1 phosphorylation. Control immunoblots were done on standard gels. **c** The LCMT1 AlphaFold-predicted structure with AKT/S6K1 phosphorylation site is indicated with a black line, and the disorder score is shown. Position and cross-species sequence conservation is shown below. **d** Cells were treated with vehicle, MK: MK-2206 (AKTi) 1 μM, Rapa: Rapamycin (mTORi) 10 μM, Torin1 (mTORi) 0.1 μM or LY: LY294002 (PI3Ki) 15 μM, for 24 h followed by immunoblotting. **e** Rapid degradation of LCMT1 in response to mitogens. Cells were either grown in complete medium or serum-starved for 24 h and then stimulated with EGF followed by immunoblotting. **f** Twenty-four hours post transfection with the indicated constructs, the cells were serum starved for 24 h and then either stimulated with EGF or continued to grow in the absence of serum for another 24 h followed by immunoblotting. **g** Forty-eight hours post transfection with the indicated constructs, protein extract was used for V5 pull-down and probed with p-AKT/S6K-substrate (p-LCMT1) or V5 antibodies. **h** S6K1 phosphorylates LCMT1. HEK293-T were transfected with V5-LCMT1 and treated with indicated drugs or siNT/siS6K1. Protein extract was used for the V5

pull-down and probed for indicated proteins. **i** S6K1 silencing restores LCMT1. HEK293-T cells were serum-starved and transfected with siS6K1 followed by EGF stimulation and then immunoblotting. **j** LCMT1 and β-TrCP interact directly. HEK293-T were transfected with constructs encoding an empty vector or FLAG-tagged FBXW7 or FLAG-tagged β-TrCP. Forty-eight hours post transfection, the cells were treated with 5 μM MLN4924 for 6 h followed by co-IP and immunoblotting for the indicated proteins. **k** HEK293-T cells were transfected with the indicated constructs. Transfected cells were then grown in the presence of 5 μM MLN4924 for 6 h followed by co-IP with anti-V5 and immunoblotting with indicated antibodies. **l** LCMT1 phosphorylation is required for its interaction with β-TrCP. HEK293-T cells were transfected with the V5-LCMT1 construct. Twenty-four hours post transfection, the cells were serum-starved for 24 h, followed by EGF stimulation for 6 h. Protein extract was used for co-IP with anti-V5 and probed for indicated proteins. **m** HEK293-T cells were transfected with si-β-TrCP followed by immunoblotting. **n** Schematic depicting S6K-induced phosphorylation and β-TrCP mediated degradation of LCMT1. All the experiments were repeated at least three times independently with similar observation.

---

markedly reduced polyubiquitination of V5-LCMT1 (Supplementary Fig. 6i), in line with the hypothesis that β-TrCP targets LCMT1 for ubiquitination and proteasomal degradation. Next, HEK293-T cells were transfected with His-ubiquitin, along with either FLAG-LCMT1, HA-β-TrCP, or both, to demonstrate direct ubiquitination of LCMT1 by β-TrCP in vivo. Pull-down of ubiquitinated proteins using Ni-NTA beads, followed by immunoblotting, revealed increased levels of polyubiquitinated FLAG-LCMT1 upon β-TrCP overexpression (Supplementary Fig. 6j). Finally, we performed in vitro ubiquitination assays. In addition to LCMT1 and S6K1, the reaction mixtures contained known cofactors (E1, E2, ubiquitin, and ATP), and β-TrCP protein. High-molecular weight species (polyubiquitinated) of LCMT1 were detected only when β-TrCP was present in the reaction mix (Supplementary Fig. 6k). Together, these data demonstrate that PI3K-AKT-mTOR-S6K1 activation leads to phosphorylation-mediated degradation of LCMT1 via β-TrCP, resulting in decreased methyl-PP2A-C sensitive PP2A holoenzyme complex formation (Fig. 5n).

*Pten* deletion in mouse prostate is associated with LCMT1 loss. The PI3K-AKT pathway is frequently altered through biallelic loss of *Pten* and activating mutations in PI3K and AKT1 in primary and metastatic prostate cancer (Supplementary Fig. 7a)[24]. Therefore, we next investigated LCMT1 stability in the context of prostate-specific *Pten* deletion. Prostate organoids generated from prostate-specific *Pten* knockout mice grew significantly faster as compared to floxed controls (Supplementary Fig. 7b) and had no detectable LCMT1 and severely decreased methyl-PP2A-C levels with a coordinate increase in p-MED1, AR, and MYC levels (Supplementary Fig. 7c). The increase in p-MED1 occurred through reduction in the binding of methyl-PP2A-C sensitive AB56αCme heterotrimer to the conserved SLiMs on the MED1 IDR. Further, RNA-seq analysis in *Pten* null organoids displayed gene-expression changes with 513 downregulated and 301 up-regulated genes compared to wild-type organoids, but interestingly no change in LCMT1 mRNA (Supplementary Fig. 7d). Accompanying LCMT1 protein loss/reduction, the GSEA analysis showed significant positive enrichment of mTORC1 and MYC gene signatures (Supplementary Fig. 7e). Since the cultured prostate glandular epithelial cell organoids from *Pten* null mice displayed significant cytological atypia with hyperactive growth and pro-survival signaling, we investigated the *Pten* null mouse prostate gland. All six-month-old (n = 9) *Pten* null prostates develop high-grade prostate intraepithelial neoplasia (HGPIN) phenotype- the main precursor lesion to adenocarcinoma[53], and they displayed a marked increase in p-MED1 staining, along with higher MYC, p-AKT and p-S6 staining - a characteristic pattern associated with LCMT1 loss/reduction (Supplementary Fig. 7f). In contrast, aged matched flox control mice (n = 7) used in the study showed no significant pathological changes. Similar to observations made in human cells, PI3K-AKT-

mTOR pathway inhibitors stabilized LCMT1 in mouse fibroblast cells (Supplementary Fig. 7g). Moreover, LCMT1 phospho-serine mutants demonstrated resistance to mitogen-induced degradation in the mouse cells, suggesting conservation of the mechanism of LCMT1 regulation across species. (Supplementary Fig. 7h). These data further support the prognostic and biological importance of LCMT1 loss in the *Pten* deletion-driven prostate cancer initiation in mice. Together, these data suggest *Pten* deletion and PI3K-AKT activation drive prostate cancer initiation and progression through degradation of LCMT1 and biased PP2A heterotrimer formation.

Restored AR activity in enzalutamide refractory prostate cancer cells is associated with reduced LCMT1. Along with PI3K pathway activation by somatic alterations, AR-mediated transcriptional addiction is a hallmark of a vast majority of refractory CRPC[24,25,54]. As first-line therapy, second-generation anti-androgens such as enzalutamide (Enza), apalutamide (Apa), and darolutamide (Daro) are efficacious in the treatment of both nonmetastatic and metastatic CRPC[27]. However, the disease invariably recurs through restoration of AR-signaling[30,54,55]. Our observation that decreased LCMT1 expression leads to increased p-AR/p-MED1, reduced sensitivity to enzalutamide, and castration-independent growth (Fig. 2), raises the possibility that restoration of AR activity observed in second-generation antiandrogen refractory CRPC[29,56] could result from AKT/S6K1-mediated LCMT1 degradation. To investigate the potential role of LCMT1 loss in persistent AR signaling in this context, we developed enzalutamide resistance (EnzaR) in LNCaP and VCaP cells by growing them continuously in the presence of the drug for close to six months and collecting the cell fractions at regular intervals. As expected, following a quiescence associated reduction in p-AR/AR and p-MED1/MED1 for the first 6-7 weeks, a steady increase in their level was evident, paralleled by a gradual loss in LCMT1 and methyl-PP2A-C, as cells evolved to attain the EnzaR state (Fig. 6a, Supplementary Fig. 8a). Though transient treatment with Enza, Apa, and Daro in the parental LNCaP cells resulted in reduced p-AR (active chromatin-bound AR) without affecting LCMT1, the LNCaP refractory derivatives displayed increased p-AR/p-MED1, accompanied by reduced LCMT1 and methyl-PP2A-C levels (Fig. 6b). Noticeably, decreased LCMT1/methyl-PP2A-C expression in EnzaR, ApaR, and DaroR cells was coupled with elevated levels of p-AKT and p-S6K1. As a result of increased phosphorylation, AR and MED1 demonstrated increased protein half-life in the enzalutamide-resistant cells (Supplementary Fig. 8b). Next, to investigate whether hyper-phosphorylated AR/MED1 as a consequence of LCMT1 loss results in a more stable AR transcriptional complex, we performed AR CoIP in parental and EnzaR cells. Interestingly, AR maintained the ability to pull-down critical coactivators such as MED1, BRD4, p300, and RNA PolII even at high salt concentrations in the EnzaR cells, compared to

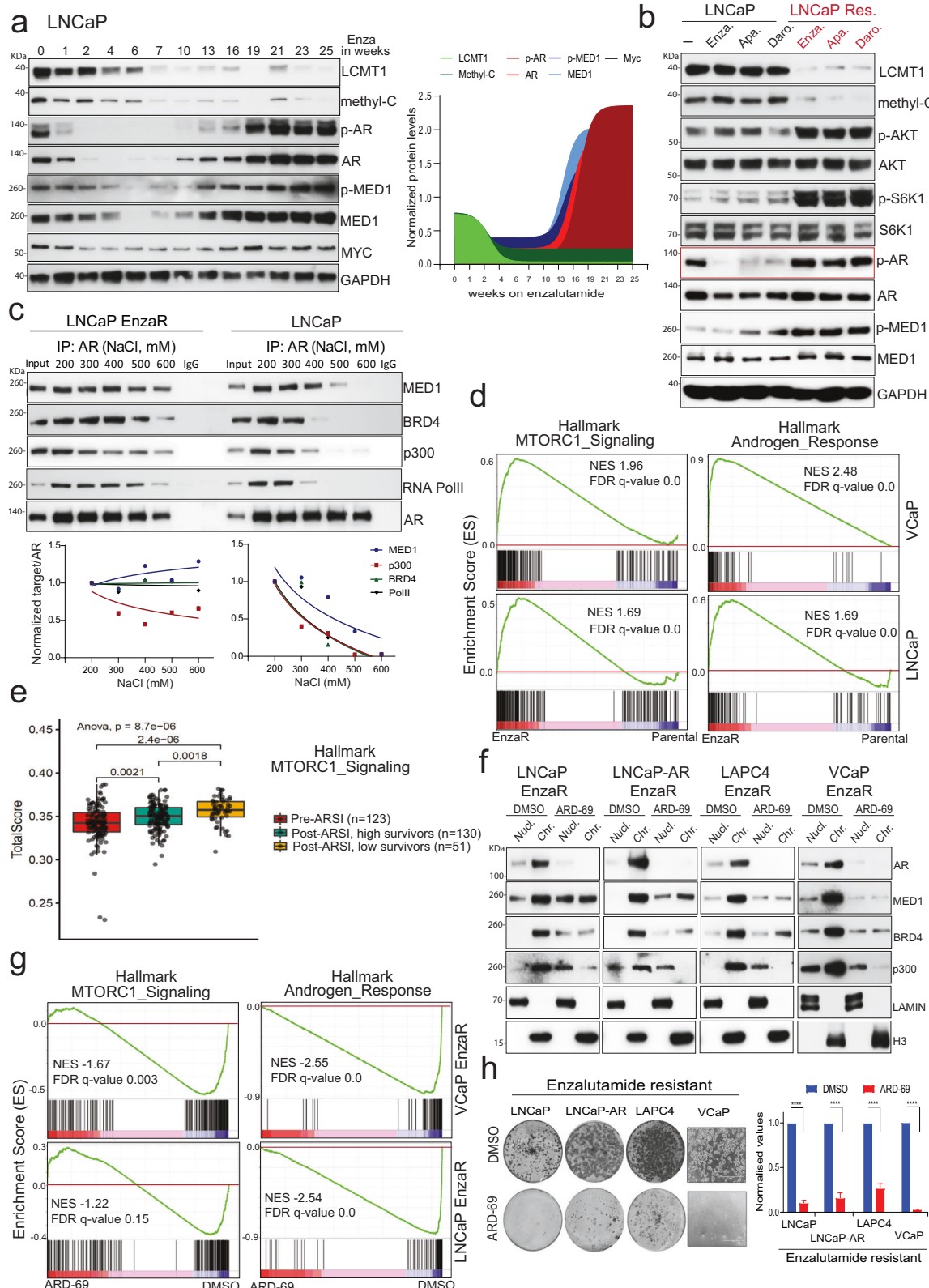

parental controls, suggestive of a stable and potentially active AR transcriptional complex (Fig. 6c). Supporting these data, RNA-seq analysis revealed significant enrichment of mTOR signaling and Androgen Response genes in VCaP- and LNCaP-EnzaR cells (Fig. 6d, Supplementary Fig. 8c), reiterating the nexus between PI3K-AKT-mTOR pathway activation and LCMT1 degradation in restoring AR signaling in enzalutamide-resistant cells. Next, we sought to determine

whether mTOR activation and AR signaling is associated with resistance or poor response to enzalutamide and other next-generation ARSI in CRPC patients. Toward this, we queried the RNA-seq dataset comprising $n = 123$ pre-ARSI and $n = 181$ post-ARSI CRPC patient samples. Though the mTOR signature was higher in post-ARSI compared to pre-ARSI CRPC tumors, interestingly, patients who had low overall survival to ARSI treatment demonstrated significantly increased mTOR

**Fig. 6 | AR addiction is associated with reduced LCMT1 and hyperactive AKT/S6K1 signaling in enzalutamide refractory (EnzaR) PCa cells. a** Evolution of EnzaR state is associated with gradual loss of LCMT1 and Leucine-309 methylation. LNCaP cells grown in the presence of enzalutamide were harvested at the indicated weeks, and the lysates prepared were immunoprobed for the indicated proteins. Normalized densitometric (ImageJ) plot for proteins is shown. The experiment was performed in two separate cell lines with the similar observations. **b** EnzaR, ApaR, and DaroR cells show reduced LCMT1 expression with a concomitant increase in p-AKT, p-S6K1, p-AR, and p-MED1. Representative immunoblots from $n = 3$ independent experiments show the effect on the indicated proteins by the transient (24 h) and chronic exposure with 10 μM of the indicated anti-androgens. **c** Top: AR co-IP was performed with increasing salt concentrations in washing buffer followed by the immunoblotting for the indicated proteins. Bar graphs show the normalized ratio of the target protein to AR. The data is representative of $n = 2$ biological replicates. **d** GSEA plot showing enrichment of the indicated msigDB signatures in

EnzaR cells. **e** Box plot showing gene signature scores for Hallmark MTORC1 signaling in pre-ARSI therapy and post-ARSI therapy stratified by overall survival from biopsy. $p$ values calculated from One-way Anova are shown. The center line shows the median, the box limits show the 75th and 25th percentiles and the whiskers show minimum-maximum values. **f** Loss of chromatin-bound AR, MED1, and BRD4 upon ARD-69 (AR-PROTAC) treatment. Chromatin and soluble nuclear fractions from cells treated with 100 nM ARD-69 for 12 h were probed for the indicated proteins. The data is representative of $n = 3$ biological replicates. **g** GSEA plot showing negative enrichment of the indicated signatures in EnzaR cells treated with ARD-69. **h** Indicated cell lines were cultured either in the presence of vehicle or 100 nM ARD-69 for 12–14 days, followed by crystal violet staining ($n = 3$). In the case of VCaP -representative bright field images are shown. Normalized quantification of crystal violet stain/cell viability is shown. Error bar represents the mean ± s.d. ($n = 3$). Statistical significance as calculated by two-tailed t-test is represented as ****$p < 0.0001$.

signature in tumors compared to patients with higher survival upon ARSI (Fig. 6e). Furthermore, the canonical AR signature did not show a positive or negative enrichment between pre- and post-ARSI tumors (Supplementary Fig. 8d), or association with time to progression (TTP) on ARSI (Supplementary Fig. 8e), suggesting continued signaling through AR due to incomplete inhibition or restoration of AR activity in these patients[34]. Next, we investigated the addiction of EnzaR cells to restored AR activity using ARD-69−a potent AR degrader (Supplementary Fig. 8f)[57]. As expected, elevated levels of chromatin-bound AR and its coactivators detected in EnzaR cells were abolished by ARD-69 (Fig. 6f), resulting in negative enrichment of Androgen Response genes (Fig. 6g, Supplementary Fig. 8g). Interestingly, genes associated with MTORC1 signaling also showed a negative enrichment upon AR degradation which support the potential crosstalk between non-genomic AR activity and PI3K-mTOR pathways[58]. Importantly, AR degradation resulted in growth inhibition of EnzaR, ApaR, and DaroR cells (Fig. 6h, Supplementary Fig. 8h). Furthermore, AR degradation induced robust apoptosis in parental as well as EnzaR derivatives of LNCaP, LAPC4, LNCaP-AR, and VCaP cells, suggesting their continued dependency on AR protein for survival (Supplementary Fig. 8i). Together, these findings suggest that restoration of a stable AR transcriptional complex as a result of AKT/S6K1-mediated LCMT1 degradation is indispensable for the development of resistance to second-generation antiandrogen targeted agents in CRPC cells.

SMAP-induced feedforward activation of LCMT1 abrogate AR addiction in enzalutamide refractory cells. Previous studies have implicated tumor suppressive PP2A heterotrimers as negative regulators of the PI3K-AKT-mTOR-S6K signaling pathway[18,20,59–61]. Targeted reactivation of specific PP2A holoenzyme has long been pursued as a potential therapy for cancer. Recently, small molecule activators of PP2A (SMAPs) engineered from phenothiazine parent compounds have shown to drive dephosphorylation of select pathogenic substrates, including AKT and S6K1[17,62]. From our finding implicating S6K1 in directly regulating LCMT1 expression resulting in stable AR transcriptional complex formation in refractory CRPC cells, we postulated that inhibition of AKT-S6K1 signaling by these PP2A modulator molecules will have a feedforward effect on LCMT1-dependent AB56αC heterotrimer assembly and AR/MED1 dephosphorylation (Fig. 7a). To test this hypothesis, we treated EnzaR, ApaR, and DaroR cells with DT-061−a SMAP that drives AB56αC heterotrimer assembly in cells, including AR driven PCa cells (Supplementary Fig. 9a)[63]. DT-061 treatment of antiandrogen refractory CRPC cells led to a decrease in p-S6K1 accompanied by elevated LCMT1/methyl-PP2A-C and lower p-AR/p-MED1 levels (Fig. 7b, Supplementary Fig. 9b). MYC, a known target of the AB56αC heterotrimer, was also downregulated in these cells upon DT-061 treatment. After observing the feedforward activation of LCMT1, we examined the effect of DT-061 on methyl-PP2A-C and AB56αC heterotrimer assembly on the chromatin and the consequent dephosphorylation of AR/MED1. As expected, compared to

the parental controls, elevated p-AR/p-MED1 and reduced AB56αC heterotrimer on chromatin was evident in the refractory cells (Fig. 7c). Treatment with DT-061 led to the accumulation of methyl-PP2A-C and AB56αC heterotrimer accompanied by lower p-AR/p-MED1 and collapsed AR transcriptional complex from the chromatin (Fig. 7c, Supplementary Fig. 9c). These observations were further confirmed using ChIP-seq analysis in EnzaR cells, where treatment with DT-061 led to a reduction in genome-wide AR and MED1 binding (Fig. 7d, Supplementary Fig. 9d). The AR degrader ARD-69 was included for comparison, which also demonstrated a robust genome-wide loss in AR and MED1 occupancy. Next, limiting our evaluation to AR and MED1 co-bound regions, we found 3382 regions enriched by AR and MED1 that were significantly depleted upon treatment with DT-061 (Supplementary Fig. 9e, f). Examples of genome browser tracks of AR target loci demonstrate the loss of AR and MED1 from the upstream regulatory regions consistent with the genome-wide findings (Fig. 7e, Supplementary Fig. 9g). Corroborating the ChIP-seq data, gene expression analysis in VCaP- and LNCaP-EnzaR cells treated with DT-061 showed repression of androgen response genes in addition to negative enrichment of MYC and PI3K-AKT-mTOR pathway genes (Fig. 7f, Supplementary Fig. 10a, b). The effect of DT-061 on AR signaling was reflected in the phenotypic response of the antiandrogen refractory cells, where SMAP treatment not only induced apoptosis but also subverted the long-term colony formation ability of these cells (Supplementary Fig. 10c, d). Next, we sought to study the in vivo efficacy of DT-061 in blocking refractory CRPC growth. Mice bearing VCaP-EnzaR tumors (~80 mm3) were randomized into three groups and treated orally with the vehicle, 15 or 50 mg/kg/twice daily DT-061 for 24 days. Treatment with DT-061 led to potent inhibition of tumor growth, triggering disease regression in more than 85% of animals (Fig. 7g, h, Supplementary Fig. 10e, f). No significant change in body weight was noted throughout DT-061 treatments, suggesting no treatment-related systemic toxicity and a broad therapeutic window (Supplementary Fig. 10g). Importantly, compared to the vehicle-treated mice, protein lysates from VCaP-EnzaR residual tumors of DT-061-treated mice showed a decrease in phosphorylated AR, MED1, and S6K1 levels and an increased LCMT1 level (Supplementary Fig. 10h). Furthermore, the B56-alpha pull-down followed by immunoblotting revealed higher levels of PP2A heterotrimers (AB56αCme) in the DT-061-treated tumor lysates (50 mg/kg) than in the vehicle group (Supplementary Fig. 10i). Together, these data demonstrate that PP2A reactivation by SMAPs such as DT-061 can restore LCMT1 through a feedforward mechanism involving inhibition of the PI3K pathway and reverse AR addiction in second-generation antiandrogen refractory CRPC (Fig. 8).

## Discussion

Hyperactivation of survival signals and the loss of negative feedback mechanisms is a hallmark of cancer. In this study, we provide insight into a detailed mechanism that underlies the loss of one such negative

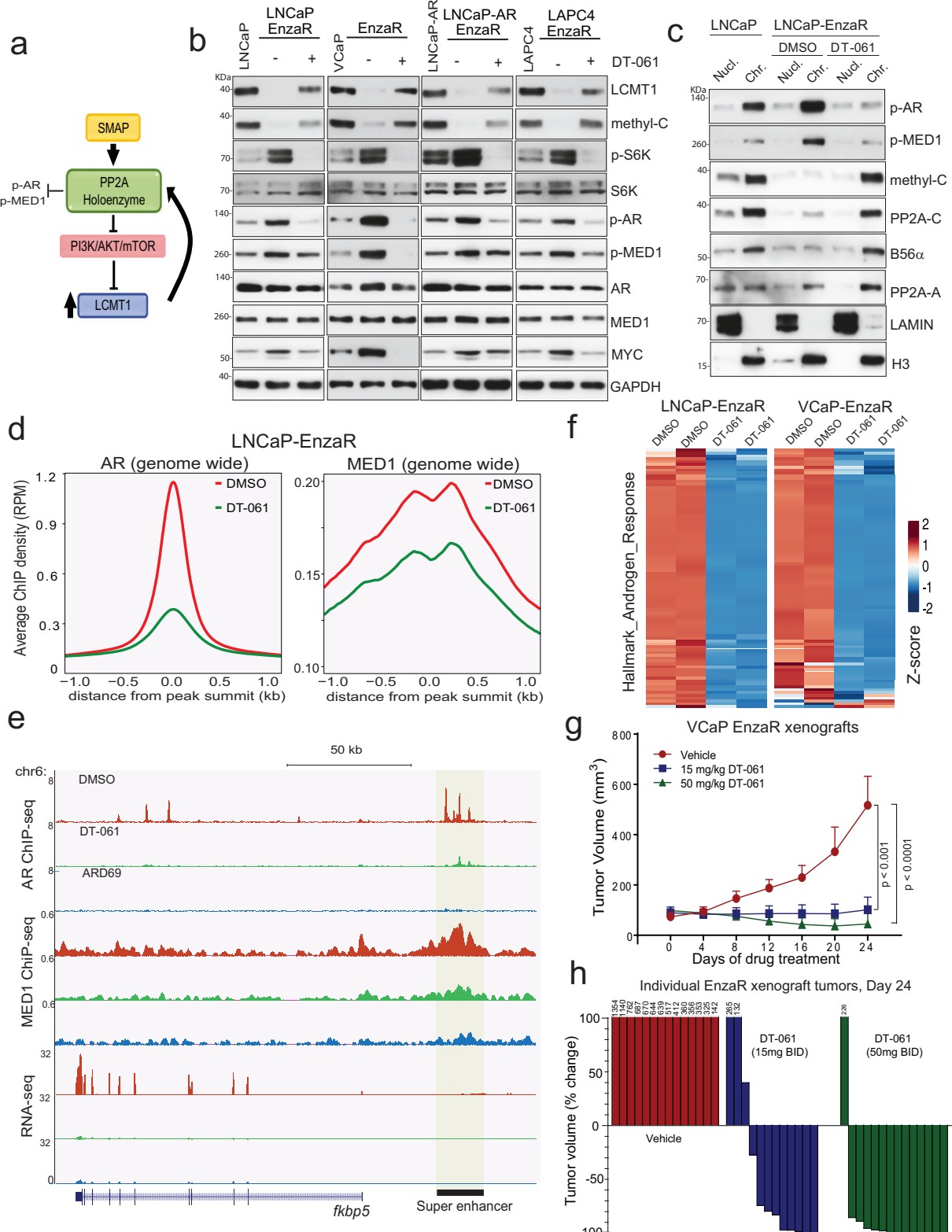

feedback control, namely the PP2A-mediated target dephosphorylation resulting in AR-addicted prostate cancer. We show that the loss of methyl-sensitive AB56αC heterotrimers, due to S6K1/β-TrCP-mediated degradation of LCMT1, results in the accumulation of hyperphosphorylated AR-MED1 complexes and restores AR signaling required for survival in next-generation ARSI-resistant CRPC cells.

Compared to a total of 428 putative serine/threonine kinases (PSKs), the human genome codes for far fewer (~50) putative serine/

threonine phosphatases (PSPs) comprising three major families: phosphoprotein phosphatases (PPPs), metal-dependent protein phosphatases (PPMs), and the aspartate-based phosphatases represented by FCP/SCP (TFIIF-associating component of RNA polymerase II CTD phosphatase/small CTD phosphatase)[3]. The observed imbalance in the existence of PSKs and PSPs coding genes led to an early misconception that these phosphatases had relatively promiscuous enzymatic activity and substrate selectivity. Significant progress has been made in the last

**Fig. 7 | SMAP induced feedforward activation of LCMT1 attenuates AR addiction in enzalutamide refractory cells. a** Schematic showing the feedforward effect of DT-061 on PP2A holoenzyme activation and LCMT1 stabilization. **b** DT-061 restores LCMT1 protein expression by reducing pS6K, resulting in lower p-AR and p-MED1 levels. Protein lysates were prepared from the EnzaR cells treated with vehicle or 10μM DT-061 for 24 h, followed by protein extraction and immunoblotting for the indicated protein. Lysates from the respective parental lines served as controls. GAPDH was used as the loading control. **c** DT-061 stabilizes the PP2A-B56α complex on the chromatin resulting in the loss of chromatin-bound p-AR and p-MED1. Chromatin and nuclear fractions from parental LNCaP and LNCaP-EnzaR cells treated with either DMSO or 10μM DT-061 for 12 h were used to probe the indicated proteins. Lamin and total H3 served as controls for the soluble nuclear and chromatin fractions, respectively. **d** DT-061 decreases genome-wide AR and MED1 binding. Genome-wide averaged AR and MED1 ChIP-seq enrichment in

LNCaP-EnzaR cells upon treatment with 10μM DT-061 for 12 h. **e** Genome browser tracks of AR and MED1 binding at *fkbp5* locus in the indicated condition for LNCaP-EnzaR cells. The superenhancer associated with this region is indicated with a black bar. The tracks at the bottom show the fkbp5 transcript expression at this corresponding locus. **f** Heatmaps of FPKM z-score displaying negative enrichment of msigDB Hallmark Androgen response signature genes in LNCaP EnzaR and VCaP-EnzaR treated with vehicle or 10 μM DT-061 for 24 h. **g** DT-061 blocks enzalutamide refractory prostate cancer growth in vivo. Mice bearing VCaP-EnzaR xenografts received vehicle (n = 12) or 15 mg/kg (n = 12) or 50 mg/kg (n = 15) DT-061 twice daily (b.i.d) for 3 weeks. Tumor volume was measured twice per week using calipers. Data are mean ± s.e.m., and *p values indicated were computed using unpaired two-tailed t-test. **h** Percent change in volume for each tumor after 24 days of treatment is shown as a waterfall plot (y-axis). For **b** and **c** the experiments were repeated at least two times independently with similar observations.

two decades concerning the biology of PPPs, and PP2A has emerged as a key tumor suppressor involved in the regulation of signal transduction, transcription, and mitosis[2,64–66]. It is generally accepted that even partial loss of PP2A phosphatase activity through differential methylation of its C subunit, resulting in biased/altered heterotrimerization, might be sufficient to unleash oncogenic signals induced by unrestrained oncogenic kinase activities[4]. However, due to its complex heterotrimer structure, evaluation of biased heterotrimerization as the readout for PP2A activity in tumor tissues has never been attempted. Here, using a highly specific mouse monoclonal antibody 7C10-C5 against methyl-L309 of PP2A-C, we provide evidence for this unique post-translational modification in prostate cancer tissues where its loss was associated with biochemical recurrence and metastasis, highlighting its potential utility as a prognostic biomarker in prostate cancer. Since AR-MED1 interacting B56α binds to the PP2A AC dimer in a LCMT1 catalyzed methyl-L309 dependent manner, our IHC data suggests the loss of AB56αC heterotrimers could directly impact AR activity, through increased accumulation of phosphorylated AR-MED1, as the disease progress from primary to more aggressive metastatic phenotype. Supporting this, multiple studies have reported increased phosphorylation of AR and MED1, and its association with poor biochemical recurrence and metastasis in PCa[29,67–70].

Moreover, AB56αC/methylation-sensitive heterotrimers negatively regulate the PI3K-AKT-mTOR-S6K signaling axis, and a high proportion of prostate cancers exhibit increased activation of this critical survival pathway[17,20,24,37,38]. Our data show a reduction in the AB56αC/methylation-sensitive heterotrimers due to phosphorylation-mediated degradation of LCMT1 involving S6K1—a downstream effector of the PI3K pathway. This strongly suggests a paradigm where persistent oncogenic PI3K signaling in cancer cells is achieved by the elimination of its negative regulator AB56αC and also other methylation-sensitive heterotrimers such as AB55C through S6K1-mediated degradation of LCMT1. Another important implication of these findings is that the PI3K signaling could also result in the inactivation of other phosphatase family members, such as PP4 and PP6, since LCMT1 is also required for the methylation of PP4-C and PP6-C catalytic subunits[21].

Methylation-sensitive PP2A heterotrimers have also been implicated in the regulation of oncogenic transcription through negative regulation of chromatin modulators and transcription factors such as BRD4, HDACs, SWI/SNF, and MYC[6,71–73]. However, though the dogma of PP2A-regulated transcription is well established, we here report the presence of canonical PP2A heterotrimer components on the chromatin. In this regard, our mechanistic data show LCMT1 catalyzed methylation-sensitive AB56αC heterotrimers on the chromatin with consequent dephosphorylation of AR-MED1, thus providing the evidence for a direct PP2A-substrate interaction on the chromatin and a consequent change in gene expression. Recently, a noncanonical PP2A holoenzyme, comprising the AC dimer and the multi-subunit RNA endonuclease integrator, Integrator–PP2A complex (INTAC), was

shown to regulate and fine-tune transcription and RNA maturation by antagonizing the action of the CDK9 kinase activity on targets like DSIF and RNAPII[64,65,74]. However, it is unclear whether the interaction between the AC dimer and Integrator complex is methylation dependent; if so, then LCMT1 could be expected to facilitate not only the canonical substrate-specific PP2A heterotrimers but also Integrator-mediated transcriptional control directly through L309 methylation of PP2A-C.

Despite recent advances in treating CRPC with second-generation ARSI such as enzalutamide, darolutamide, and apalutamide, de novo resistance is observed in 20–40% of patients, and most patients invariably acquire resistance to these therapies[28,30]. Though restoration of AR signaling through AR point mutations, overexpression, genomic amplification, and constitutively active splice variants is shown to promote resistance to AR antagonism, the underlying molecular basis for AR transcriptional activity and its requirement for the survival of ARSI-resistant cells is unclear[56]. The findings here implicate AKT/S6K1-mediated LCMT1 degradation as key to developing resistance to ARSI in CRPC cells that continue to rely on AR activity for survival. Furthermore, our data showing the sensitivity to AR degraders in the ARSI-resistant cells illustrate the incomplete targeting of AR by enzalutamide, apalutamide, or darolutamide. Therefore, further targeting of the AR through PROTAC-based therapies that are under investigation[75], may be clinically beneficial in patients who develop resistance to next-generation ARSI. Furthermore, the loss of methylation-sensitive AB56αC heterotrimers, and potentially AB55C heterotrimers which is known to negatively regulate AKT[76], leads to increased PI3K/AKT signaling in ARSI refractory CRPC cells. This suggests that drugs targeting PI3K/AKT could be beneficial in treating ARSI refractory CRPC.

Beyond their biological significance, the results presented in this work strongly support the use of SMAPs in treating ARSI-resistant CRPC. The SMAP induced biased PP2A reactivation coupled with feedforward activation of LCMT1 through AKT/S6K1 inhibition provide an excellent opportunity for targeted eviction of the hyperphosphorylated AR transcriptional complex from chromatin. While target-specific molecular glues like SMAPs are still in their early phase of development, our study suggests that combining PI3K/AKT/S6K1 pathway inhibitors with antiandrogen therapies cannot be ruled out in AR-addicted prostate cancer. Furthermore, the mechanistic insights presented in this work indicate that direct stabilization of LCMT1 by phosphorylation-targeting chimeras (PhosTACs)[77] based therapeutics could provide additional opportunity for targeted activation of methyl-sensitive biased PP2A heterotrimers in treating transcription addicted cancers.

## Methods

### Mice
Animal studies were approved by the Institutional Animal Care and Use Committee (IACUC) at The University of Pennsylvania and/or The

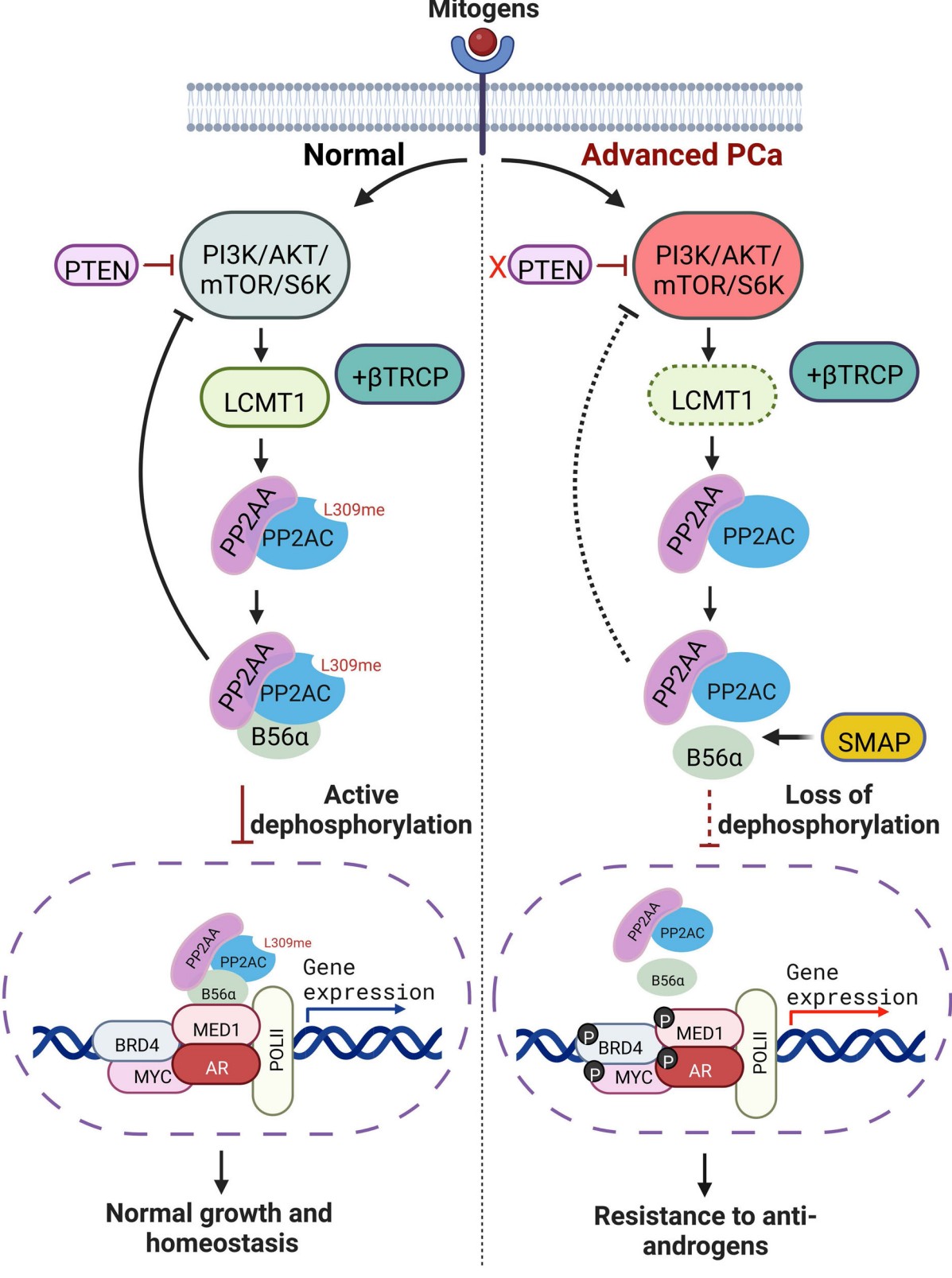

**Fig. 8 | Model showing the S6K-LCMT1-PP2A-AR axis in normal prostate and advanced refractory prostate cancer.** Methyl-sensitive PP2A holoenzyme complex is destabilized by PI3K/AKT/mTOR pathway-mediated degradation of LCMT1 through β-TRCP, resulting in hyperphosphorylated chromatin-bound stable AR transcriptional complex. Biased heterotrimerization of PP2A by small molecular activators of phosphatase lead to a feedforward LCMT1 stabilization via AKT-S6K1 inhibition, resulting in dephosphorylation of the AR transcriptional complex and death of refractory AR-addicted cells. The schematic was created with BioRender.com.

University of Michigan. Animal use and care was in strict compliance with institutional guidelines and all experiments conformed to the relevant regulatory standards by the Universities. The maximal tumor size in our study models (mice) never exceeded 1 cm in diameter as allowed by the above ethics committee. NOD SCID or NCI SCID/NCr athymic nude mice were obtained from the Jackson Laboratory (strain code: 005557) and Charles River (strain code: 561). Prostate-specific *Pten* knockout mice were generated by crossing Pbsn-Cre male mice Tg (Pbsn-cre) 4Prb/J (Jackson laboratory, Strain: 026662) with Pten floxed mice (B6.129S4-Ptentm1Hwu/J, Strain: 006440). All mice were housed in a pathogen-free animal barrier facility and all in vivo experiments were initiated with male mice aged 5-8 weeks.

The maintenance of mice and the experimental procedures (immunizations) have been conducted according to the Austrian Animal Experiments Act and have been approved by the Austrian Federal Ministry of Science and Research BMWFW-66.009/0211-WF/V/3b/2015 and the animal experiments ethics committee of the Medical University of Vienna.

### Cell cultures

The cell lines used in this study were obtained from American type cell culture (ATCC) unless mentioned. LNCaP-AR cells were a generous gift from Dr. Charles Sawyers- Memorial Sloan-Kettering Cancer Center, New York, NY. LNCaP, LNCaP-AR, and LAPC4 prostate cancer cell lines were grown in RPMI 1640 (Gibco, 11875093), VCaP- prostate cancer line, HEK293-T, and NIH-3T3, cells were grown in DMEM with Glutamax (Gibco, 21013024). HAP1 wild-type and LCMT1 CRISPR Cas9 deletion cells (Horizon Discovery, C631, HZGHC004373c001) were grown in Iscove's Modified Dulbecco's Medium (IMDM, Life Technologies, 12440053). The medium was supplemented with 10% of FBS (HYC, SH30910.03) or FCS (Sigma, F7524) and 1% of Penicillin Streptomycin Solution (Invitrogen, 15140122). LNCaP-AR enzalutamide-resistant cell lines were derived from enzalutamide-resistant tumor xenografts as described previously[78]. LNCaP, LNCaP-AR, VCaP, and LAPC4 enzalutamide-resistant cell lines were generated by culturing their parental lines in presence of enzalutamide (10 μM) for 4–6 months in vitro. Surviving enzalutamide-resistant polyclonal pools were maintained in 10 μM enzalutamide throughout. Apalutamide- and darolutamide-resistant LNCaP, LNCaP-AR, VCaP, and LAPC4 lines were generated by culturing the cells in 10 μM of the respective drugs for at least two months.

All experiments using antiandrogen-resistant pools were done in the presence of enzalutamide, except when cells were treated with AR-PROTAC ARD-69. shLCMT1 or non-targeting control shNTC cell lines were created transducing shLCMT1-GFP or shNTC-GFP lentiviral vectors in LNCaP, VCaP, LAPC4, and LNCaP-AR cells. Cells were sorted for GFP expression. Knockdown was confirmed by immunoblotting. LCMT1 overexpression or control cell lines were generated by transducing pLX304-LacZ or pLX304-LCMT1 in LNCaP cells. Cells were selected using Blasticidin. Overexpression was confirmed by immunoblot. All the cell lines were routinely tested for mycoplasma contamination using the MycoAlert Mycoplasma Detection kit (Lonza, T07-418) and were maintained in a humidified incubator at 37 °C and 5% CO₂. The complete list of cell lines used is indicated in Supplementary Data 1.

### Reagents and kits

The complete list of the reagents used throughout the study and their commercial suppliers are given in Supplementary Data 1.

### Patient tissues

A set of tissue microarray (TMA) slides (section thickness 3 μm) containing a single 0.6 mm tissue punch from each of 17,747 patients was constructed at the Institute of Pathology, University Medical Center Hamburg-Eppendorf, Germany. The composition of the TMA is given in Supplementary Table 1. Further details of the TMA were described previously[79].

### Immunohistochemistry for methyl-L309 PP2A-C

Immunohistochemistry was performed on a DAKO autostainer. TMA slides were deparaffinized and rehydrated in a graduated series of ethanol. Slides were subjected to epitope retrieval using 10 mM Tris−HCl buffer pH9.0 containing 1 mM EDTA. The slides were then subjected to H₂O₂ and protein exposure to remove non-specific binding. The slides were exposed overnight to the antibody (MeC-PP2a, mouse monoclonal antibody, clone 7C10, 1:200). After washing with TBST. The slides were exposed to DAKO Envision+ (Goat anti-mouse polymerized HRP) and subsequently the chromogen, diaminobenzidine (DAB). After counterstaining with hematoxylin, the slides were digitally scanned using an APERIO scanner. In order to determine the specificity of the antibody, the antibody was pre-incubated with 10x molar excess of the methylated and unmodified peptides prior to incubation with slides.

The digital slides were examined using QuPath and the H-score and Allred scores were submitted back to the originator of the TMA slides to be correlated with the clinical data.

### ELISA

ELISA 96-well plates (Thermo Scientific, Medisorb) were coated with 50 μl peptide solution (2 μg/ml in TBS) at 4 °C over night. The plate was blocked with 2% BSA in TBS for 1 h at RT and incubated with primary antibodies in TBS for 1 to 2 h at room temperature. Incubation with secondary peroxidase conjugated antimouse was performed for 1 h at RT followed by detection with TMB (3′,5,5′,5′-tetramethylbenzidine; Sigma, Cat T2885) and H₂O₂ in a sodium acetate buffer pH = 6. The reaction was stopped by the addition of 1 N H₂SO₄ and the absorbance was measured at 450 nm, for background correction the absorption of 560 nm was subtracted. For ELISA quantification the signals of the antibodies were normalized to the signal of the cognate target peptide of the antibody which was artificially set to 1.

### Affinity measurements

Measurements were performed using Biacore T200 instrument. The sensorgrams obtained were analyzed with Biacore T200 Evaluation Software (version 3.1). Flow cells were coated with 30 μg/mL antimouse IgG antibody (in immobilization buffer) using the mouse antibody capture kit (GE Healthcare, BR-1008-38, Lot 10294578) via amine coupling according to the manufacturer's protocol. The immobilization levels obtained were 11605.9 and 11333.9 RU. The antibody (7C10) was applied with a concentration of 50 μg/ml and reached a response level of approx. 1200 RU. The antigens were used in different concentrations:

PP2A Me 5 nM, 10 nM, 20 nM, 40 nM, 80 nM and 160 nM for 7C10
PP4 Me 50 nM, 100 nM, 200 nM, 400 nM, 800 nM for 7C10

Antibody capturing time was adjusted to reach approx. 1200 RU, antigen association time was 360 s, dissociation time was 450 s. The flow rate was 40 μl/min. Regeneration of the chip was performed injecting 10 mM glycine-HCl pH 1.7 for 180 s with a flow rate of 20 μl/min. The value of the equilibrium dissociation constant (KD) was obtained by fitting a plot of response at equilibrium (Req) against the respective concentration of the analyte (steady state analysis).

**Peptides.** Peptides were purchased from Pichem. The sequences of the peptides used in this study are as follows:

Leu[309]: Ac-His-Val-Thr-Arg-Arg-Thr-Pro-Asp-Tyr-Phe-Leu-OH
meLeu[309]: Ac-His-Val-Thr-Arg-Arg-Thr-Pro-Asp-Tyr-Phe-Leu-OMe
amLeu[309]: Ac-His-Val-Thr-Arg-Arg-Thr-Pro-Asp-Tyr-Phe-Leu-NH₂
Leu[307]: Ac-Ile-Pro-Ser-Lys-Lys-Pro-Val-Ala-Asp-Tyr-Phe-Leu-OH
meLeu[307]: Ac-Ile-Pro-Ser-Lys-Lys-Pro-Val-Ala-Asp-Tyr-Phe-Leu-OMe
amLeu[307]: Ac-Ile-Pro-Ser-Lys-Lys-Pro-Val-Ala-Asp-Tyr-Phe-Leu-NH₂

Leu[305]: Ac-Ile-Pro-Pro-Arg-Thr-Thr-Thr-Pro-Tyr-Phe-Leu-OH
meLeu[305]: Ac-Ile-Pro-Pro-Arg-Thr-Thr-Thr-Pro-Tyr-Phe-Leu-OMe
amLeu[305]: Ac-Ile-Pro-Pro-Arg-Thr-Thr-Thr-Pro-Tyr-Phe-Leu-NH₂

## Co-immunoprecipitations and immunoblotting

For immunoblotting (IB), cells were harvested and lysed by sonication in RIPA buffer (Boston Bioproducts, BP-115DG)- (50 mM Tris 7.4, 150 mM NaCl, 1% NP40 0.1 % SDS, 0.5% Sodium Deoxycholate, 1 mM EDTA, 5 mM EGTA), supplemented with protease inhibitor cocktail (Pierce, A32965) and phosphatase inhibitor (Thermo Scientific, 1861280). Protein concentration was determined by the standard BCA method, and an equal amount of protein from each sample was boiled in a sample buffer and subjected to SDS−PAGE. The proteins were transferred to PVDF membrane (Millipore, IPVH00010) or nitrocellu-lose membranes, blocked with PBS-T (Sigma, 274348)- 5% (w/v) nonfat milk in PBS containing 0.1% Tween-20, probed with the relevant anti-bodies for 2 h at room temperature or overnight at 4 °C, and subse-quently washed and probed with species-specific secondary antibodies coupled to horseradish peroxidase (Kindle Biosciences, R1005 and R1006) at room temperature for 2 h. For Methyl-PP2A-C protein detection, IP lysis buffer (20 mM Tris pH 7.5, 150 mM NaCl, 0.5% Triton-X 100, Protease/Phosphatase Inhibitor) was used instead of RIPA buffer, and the antibodies were probed in 0.5% nonfat milk in PBS-T. Immunoreactive proteins were detected by an enhanced che-miluminescence system as per the manufacturer's protocol (GE Healthcare) or Kwik Quant Imager (Kindle Biosciences).

For immunoprecipitation (IP) experiments, nuclear or whole cell protein extracts were obtained from cells using NE-PER nuclear extraction kit (Thermo Scientific, 78835) or RIPA buffer, respectively. The nuclear pellet was then lysed in an IP buffer by sonication. Nuclear lysates (0.5–2.0 mg) were pre-cleaned by incubation with protein G Dynabeads (Life Technologies, 10003D or Thermo Scientific, 14321D) for 1 h on a rotator at 4 °C. Next, an antibody (2–5 µg) was added to the pre-cleared lysates and incubated on a rotator at 4 °C overnight. The following day, Protein G Dynabeads were added to the nuclear lysate-antibody mix for 1 h. Beads were washed twice in IP or RIPA buffer containing 300 mM NaCl, resuspended in 40 µL of 2× loading buffer, boiled at 100 °C for 5 min, and subjected to SDS−PAGE and immuno-blotting. For FLAG-tag immunoprecipitation, the cells were lysed in NP40 buffer (0.1% NP40, 15 mM Tris−HCl pH 7.4, 1 mM EDTA, 150 mM NaCl, 1 mM MgCl₂, 10% Glycerol) containing protease and phosphatase inhibitors and the lysates were incubated with anti-FLAG M2 beads for 2 h at 4 °C or used for V5 antibody pull-down assay. For B56a and B55a/d immunoprecipitations one to two 15 cm cell culture dishes with approximately 70% confluent cells or tumor tissues were lysed in lysis buffer (10% (v/v) glycerol; 135 mM NaCl; 20 mM Tris, pH 8.0; 1% Non-idet P-40; 1 mM PMSF; 0.03 U/ml aprotinin (Sigma), 1× Complete (Roche)) by 2″ sonication, centrifuged at 10956 × g at 4 °C for 10 min. Supernatant of lysates was adjusted to a protein concentration of 2 to 2.5 µg/µl. 1/10 of the lysate (input) was boiled in protein sample buffer and 1 ml of lysate was incubated either with B56α (1D3-E12) or B55α/δ (2G9) or B56α (ab89621) antibody crosslinked to protein G-Sepharose beads (GE-Healthcare) to immunoprecipitate endogenous B55α/δ (clone 2G9) or B56α (clone 1D3-E12) for 1 h. The immune complexes were washed once with lysis buffer, 3× with Tris-buffered saline (25 mM Tris, 135 mM NaCl, 2.6 mM KCL pH = 7.4 with HCl). For PP2A-A-V5 coimmunoprecipitation experiments, cells were harvested, and coim-munoprecipitation was performed per Dynabeads CoImmunopreci-pitation protocol (ThermoFisher, 14321D). V5 antibody (Bio-Rad) was coupled at a concentration of 7 µg/mg of Dynabeads. Cell lysate was incubated with the Dynabeads for 30 min, rotating, at 4 °C. Fresh conjugated beads were prepared for each biological replicates. The immunoprecipitate was boiled for 5 min at 95 °C in protein sample buffer. After that, SDS−PAGE, and immunoblotting was performed as described above. Densitometric quantification was performed using the Image J software and Image Lab Software.

All experiments were performed at least three times. Statistical significance of immunoblotting data was assessed using a one-way analysis of variance (ANOVA) and a Tukey's honestly significantly dif-ferent (HSD) post hoc test for multiple comparisons, or unpaired Students t-test for pairwise comparisons. In all cases, P values of <0.05 were considered to be statistically significant and are indicated with *P < 0.05, **P < 0.01, ***P < 0.005, ****P < 0.001. Statistical analysis was performed in GraphPad Prism.

The antibodies used in this study are given in Supplementary Data 1. All antibodies were employed at dilutions suggested by the manufacturer or as standardized.

## Drug preparations

CDK7 inhibitor-THZ1 (MedChemExpress, HY-80013A/CS-3168), R1881C-III (Sigma, R0908), Trametinib (Selleckchem, S2673), Enzalu-tamide (Selleckchem, S1250), Apalutamide (Selleckchem, S2840), Darolutamide (Selleckchem, S7559), MLN4924 (Sigma, 5.05477), Torin1 (Sigma, 475991), LY294002 (MedChemExpress, HY-10108), MK-2206 (MedChemExpress, HY-10358), MG132 (MedChemExpress, HY-13259), DT-061 (MedChemExpress, HY-112929), Cyclohexi-mide (Sigma, 66-81-9) and Rapamycin (Sigma, 37094) were dissolved and aliquoted in DMSO (Sigma, D2650). EGF (Abcam, ab259398) was dis-solved and aliquoted in water.

## EGF activation

For Epidermal Growth Factor (EGF) (Thermo Scientific, PHG0311L) stimulation, the cells were plated in complete media at 60–70% con-fluency. After 24 h cells were washed with PBS, serum starved for 24 h and then treated with 100 nM EGF for the desired time points. Simi-larly, for androgen starvation, the cells were cultured in a medium without Phenol Red (Invitrogen, 11835030) and 10% charcoal-dextran stripped FBS (Gemini Bio-Products, 100-119) for the required time points post 24 h growth in a regular medium.

## RNA interference loss of function studies

For gene knockdown experiments, cells were seeded in six-well plates and transfected with 100–200 nM ON-TARGET plus SMART pool siRNA targeting S6K (Dharmacon, L-003616-00-0005), β-TrCP (Dhar-macon, L-003463-00-0005) and non-targeting pool as a negative control (Dharmacon, D-001810-10-05) using lipofectamine RNAiMAX (Invitrogen, 13778150) according to the manufacturer's instructions. The cells were then harvested 48/72 h post-transfection and used for the data presented.

## Proliferation and colony formation assays

For the cell proliferation assay, the cells were seeded at a density of 50,000 per well in 12-well plates (n = 3). Twenty-four hours after seeding the cells were treated with ARD-69 or Enzalutamide or DT-061 or DMSO. Medium with the drug/compound was changed after every 48 h. The cells were harvested, and live cell numbers were counted in the conditions used by Countess II FL (Invitrogen, AMQAF1000).

For the colony formation assay, approximately 10,000 cells/well in six-well plates (n = 3) were seeded and treated with the required drugs/compounds or vehicle for 12-14 days. Media was replenished every 3–4 days. Colonies were fixed and stained using 0.5% (w/v) crystal violet (Sigma, C0775) in 20% (v/v) methanol for 30 min, washed with distilled deionized water, and air-dried. After scanning the plate, the stained wells were destained with 500 µL 10% acetic acid and the absorbance was determined at 590 nm using a spectrophotometer (Synergy HT, BioTek Instruments, Vermont-USA). For VCaP or VCaP-Enzalutamide-resistant cells, viability was assessed by Cell-Titer GLO (Promega, PR G7571).

## Cellular protein fraction assays

Chromatin-bound proteins were extracted following a protocol previously described[29]. In brief, 10 million cells were collected, washed with DPBS, and resuspended in 250 µl Buffer A (10 mM HEPES pH 7.9, 10 mM KCl, 1.5 mM MgCl$_2$, 0.34 M sucrose, 10% glycerol, 1 mM DTT) supplemented with 0.1% Triton-X-100. After incubation on ice for 10 min, the nuclear pellet was collected by centrifugation at 1300 × $g$ for 5 min at 4 °C, washed in Buffer A, and resuspended in Buffer B (3 mM EDTA, 0.2 mM EGTA, 1 mM DTT) with the same centrifugation settings, and incubated on ice for 30 min. The chromatin pellet was collected by centrifugation at 1700 × $g$ for 5 min at 4 °C, washed and resuspended in Buffer B with 150 mM NaCl, and incubated on ice for 20 min. After centrifugation at 1700 × $g$ for 5 min to remove proteins soluble in 150 mM salt concentrations, the pellet was then incubated in Buffer B with 300 mM NaCl on ice for 20 min and centrifuged again at 1700 × $g$ to obtain the final chromatin pellet. The chromatin pellet was dissolved in a sample buffer, sonicated for 15 s, and boiled at 95 °C for 10 min. Immunoblot analysis was conducted on samples as described previously. All buffers were supplemented with Pierce protease inhibitor and Halt protease & phosphatase inhibitors.

## Quantitative real-time PCR

The miRNeasy Mini Kit (Qiagen, 74106) was used to isolate total RNA from cells and the SuperScriptIV (Life Technologies, 18090200) kit was used to synthesize cDNA from 1 µg total RNA. qRT-PCR was performed using Fast SYBR Master Mix (Life Technologies, 4385617) or Taqman Fast Advance MMIX (Life Technologies, 4444964), and analyzed on QuantStudio3 (Applied Biosystems, USA). The ΔΔCt method was used to quantify target mRNA expression, as normalized to GAPDH transcript levels. Primers were designed using Primer3 Input (version 0.4.0) (http://bioinfo.ut.ee/primer3-0.4.0/primer3) and synthesized by Integrated DNA Technologies. The primer sequences for the SYBR green assays qPCR can be found in the Supplementary Data 1.

## Immunofluorescence microscopy

Cells seeded on coverslips were fixed with 4% paraformaldehyde for 20 min followed by washing with PBS, permeabilized with PBS containing 0.5% Triton-X 100, and blocked in 3% BSA for 1 h at room temperature. After blocking, cells were incubated overnight at 4 °C with primary antibodies for pMED1 (1:500), pAR (1:100), and methyl C (1:50). Coverslips were washed three times with PBS-T and incubated with the respective Alexafluor secondary antibodies (1:200) for 1 h at room temperature followed by counterstaining with DAPI (Sigma). Coverslips were mounted on glass slides using VECTASHIELD PLUS Antifade Mounting Medium (Vector Laboratories) and cured overnight. Images were captured on a Zeiss LSM 880 confocal microscope. FIJI software was used to integrate the signal for quantitative image analysis. For specific quantification of nuclear staining, regions of interest were overlaid with the DAPI signal, and only co-localizing regions were included in the integration to exclude background cytoplasmic signals.

## Phos-Tag SDS/PAGE study

The protein samples were resolved on 6% PAGE gel containing 10 µM Phos-tag acrylamide AAL-107 (FUJIFILM Wako Pure Chemical Corp, 30493521) and 50 µM MnCl$_2$ according to the manufacturer's instructions. The Phos-tag gels were washed with transfer buffer containing 4 mM EDTA for 20 min and then were washed with transfer buffer containing 0.1% SDS for 5 min before transfer onto PVDF membranes. The membranes were then immunoblotted with specific primary antibodies overnight followed by imaging and analysis.

## B56α-MED1 protein interaction study

Glutathione Sepharose beads (GE Healthcare, 17-0756-01) were equilibrated by incubating for 10 min (3×) in PBS (Gibco, 10010023)-137 mM NaCl, 2.7 mM KCl, 8 mM Na$_2$HPO$_4$, and 2 mM KH$_2$PO$_4$ at 4 °C. Upon equilibration purified GST-tagged protein was loaded on the beads by incubating at room temperature for 45 min on a shaker. Following that 2–3 mg of nuclear protein, lysate was incubated with protein-bound glutathione beads at 4 °C for overnight on a shaker. The next day, the bead-bound complex was washed four times with PBS to remove unbound protein, and the bound proteins were eluted from beads by boiling the sample in SDS loading dye at 100 °C. Elute was resolved on SDS–PAGE and proteins of interest were detected using immunoblotting.

## Ubiquitination assays

For in vivo ubiquitylation assay, HA- β-TrCP1, Flag-LCMT1, and His-ubiquitin expression plasmids were co-transfected into HEK293-T cells. After 36 h, 5 µM MG132 (Selleckchem, S2619) was added for 6 h before harvesting the cells. Transfected cells were lysed with 1% SDS and sonicated. The cell lysates were diluted with NP10 buffer and incubated with Ni-NTA beads at 4 °C for 2 h. Next, beads were washed three times with the NP40 buffer, boiled in 50 µl of 2× SDS loading buffer for 5 min, followed by western blot analysis with anti-V5-LCMT1 antibody to detect polyubiquitination.

For in vitro ubiquitylation assay, the ubiquitylation of LCMT1 was performed in a volume of 10 µl, containing 50 mM Tris pH 7.6, 5 mM MgCl2, 2 mM ATP, 1.5 ng µl−1 E1, 10 ng µl−1 Ubc3, 10 ng µl−1 Ubc5, 2.5 µg µl−1 ubiquitin (Boston Biochem), S6 kinase, and 1 µl of unlabeled in vitro transcribed/translated LCMT1 and/or βTrCP1. The reactions were incubated at 30 °C for 1 h, subjected to SDS–PAGE, then analyzed by immunoblotting with an anti-LCMT1 antibody.

## Plasmids and lentivirus

The pLX304-V5-LCMT1-WT plasmid was purchased from DNASU (HsCD00441432) and pLX304-V5-LCMT1-S8A, pLX304-V5-LCMT1-S9A, pLX304-V5-LCMT1-S12A, pLX304-V5-LCMT1-S8.9.12 A constructs were created from it through Site-directed mutagenesis. Similarly, pWZL hygro FLAG-HA TRAP220-WT (Addgene, 17433) was used to create pWZL-HA-MED1-2A-a, pWZL-HA-MED1-2A-b, and pWZL-HA-MED1-2A-ab plasmids. Primers, shown in the Supplementary Data 1, were designed using the Agilent primer design program (https://www.agilent.com/store/primerDesignProgram.jsp), and Site-directed mutagenesis was performed using the QuikChange II XL site-directed Mutagenesis Kit (Agilent Technologies, 200521) as per the manufacturer's protocol. All the mutations were verified by Sanger sequencing.

Other plasmids used in this study are V245 pCEP-4HA B56α (Addgene, 14532), PPP2R1A plasmid in pLX304 (DNASU, HsCD00444402), Flag-FBXW7, Flag- β-TrCP, pFN21K-AR-Halo, and MSCV-myrAkt (Addgene plasmid_ 49267-deposited by Lawrence Kane & Arthur Weiss), pLX304-LacZ (Addgene, 42560), pLX304-EGFP (described previously by O'Connor et al., Oncogene 2020), shNTC and shLCMT1 plasmids (Origene, TL317193). The plasmids were transfected into HEK293-T or LNCaP cells using Lipofectamine 2000 (Invitrogen Life Technologies, 12566014) as per the manufacturer's protocol. The cells were then harvested 48/72 h post transfection and subjected to immunoprecipitation or immunoblotting or used for Lentivirus production. Lentiviral particles were generated in the lab at the University of Pennsylvania (UPenn) or in collaboration with the Vector Core at the University of Michigan. Following lentiviral production, viral particles were incubated on cells in penicillin/streptomycin-free media for 24 h, followed by media replacement with normal media. After 72 h, cells were selected in 16 µg/mL Blasticidin-Invivogen (pLX304 constructs) or sorted by FACS for GFP positivity (shLCMT constructs) to generate stable cell lines. Knockdown or overexpression of proteins was confirmed by immunoblotting.

## RNA-seq library preparation and sequencing

RNA-seq libraries were constructed using the TruSeq sample Prep Kit V2.5 (Illumina) according to the manufacturer's instructions. In brief, 1 µg of purified RNA was poly-A selected and fragmented with a fragmentation enzyme. Poly-A selected/fragmented RNA templates were used to synthesize the first and second strands, and end-repair to PCR amplification was performed according to the manufacturer's protocol. Libraries were purified and validated for appropriate size on a 2100 Bioanalyzer DNA 1000 chip (Agilent Technologies, Inc.). DNA library concentration was assessed using Qubit and normalized to 4 nM before pooling. Libraries were pooled in an equimolar fashion and diluted to a loading concentration of 1.8 pM. Library pools were clustered and run on the Nextseq500 platform (Illumina Inc.) with single-end reads of 75 bases.

## RNA-seq analysis

Single-end reads were demultiplexed using Illumina's bcl2fastq CLI tool. Fastqc was used to quality check the fastq files. Reads were then aligned using STAR v2.7.3.a[80] with default parameters to either the HG19 or MM10 reference genomes. To remove low-quality/duplicate reads and sort the aligned BAM files, Samtools (v1.13) was used. Filtered BAM files were converted to BigWig files to view on the UCSC browser using bamCoverage (v3.5.1) from the Deeptools suite. Cufflinks, Cuffquant, and Cuffnorm v2.2.1[81] were used to assemble, quantify, and normalize transcripts (FPKM) to assemble the count tables, respectively.

## Differential gene-expression analysis

Differential expression analysis was done using the Bioconductor package, DESeq2 (https://bioconductor.org/packages/release/bioc/html/DESeq.html). DESeq2[82] performs internal normalization to account for library size and RNA composition bias provided the raw counts. Genes with a fold change of at least 1 and a $q$-value < 0.05 were considered to be differentially expressed.

## Gene set enrichment analysis

Gene enrichment analysis or GSEA is a tool that determines over-represented genes in predefined gene sets that are biologically related using statistical methods[83]. Comparisons were done between Parental vs Enzalutamide-Resistant cell lines (LNCAP, VCAP), LCMT1 over-expressing vs LAZ cells (LNCAP), and PTEN vs PTEN-Null (mouse organoids) using Hallmark signatures that are available in the Molecular Signatures Database (MSigDB) by BROAD institute.

## Chromatin Immunoprecipitation and Library preparation

Chromatin immunoprecipitation (ChIP) was performed using the iDeal ChIP-seq kit for Transcription Factors (Diagenode, C01010170) according to the manufacturer's instructions. In brief, cells were collected and subjected to DNA-protein cross-linking using formaldehyde. After cell lysis, the cross-linked chromatin was fragmented and immunoprecipitated using Protein A magnetic beads and antibodies targeting transcription factors of interest. IPure magnetic beads were used to purify eluted DNA, which then underwent library preparation and sequencing. Library preparation for Illumina NextSeq was performed using the NEBNext Ultra II DNA library prep kit (NEB) as per the manufacturer's protocol. The quality of the library prepared was assayed using a BioAnalyzer-2100 (Agilent, G2939BA) and quantified using Qubit prior to loading of samples onto an Illumina NextSeq-500 to perform sequencing of 75 nucleotide paired-end reads at 40 million read depth.

## ChIP-seq analysis

Paired-end reads were demultiplexed using Illumina's bcl2fastq CLI tool. Fastqc was used to quality check the fastq files. Reads were then aligned using Bowtie2 (v2.2.5) with default parameters to the HG19

reference genome. MarkDuplicates from the genome analysis toolkit (GATK, v4.2.3) was used to mark duplicates. To remove low-quality/duplicate reads and sort the aligned BAM files, Samtools (v1.13) was used. Filtered BAM files were converted to BigWig files to view on the UCSC browser using bamCoverage (v3.5.1) from the Deeptools suite. MACS2 (v2.2.6) was used to compute and find enriched peaks in BAM files[84]. To overlap peaks and remove problematic regions known to produce enriched signals in several next-generation sequencing experiments, known as blacklists (https://github.com/Boyle-Lab/Blacklist), bedtools intersect (v2.30.0) was used.

## Pre-ARSI and post-ARSI CRPC data analysis

From 4/2005-7/2021, 268 prostate cancer patients older than 18 years of age underwent tissue collection for tumoral RNA-sequencing at the University of Michigan (HUM00046018, HUM00048105, HUM00067928, SU2C). Clinical data including information on ARSI treatment regimens (abiraterone, enzalutamide, apalutamide, darolutamide) and survival was collected from 05/2021-01/2022. RNA-sequencing data for 313 RNA libraries across 290 biopsies was processed using Turnkey Precision Oncology, a flexible in house computational pipeline. Gene signatures for androgen response and mTORC1 signaling were obtained from the Molecular Signatures Database[83]. Gene signature scores were computed using singScore[85]. Samples were split into those pre-/post-ARSI. The post-ARSI patients were further split based on high/low survival from the date of biopsy, and high/low time to progression on ARSI. Analysis of variance (Anova) and Wilcoxon signed-rank tests were used to determine differences between groups.

## *Pten* knockout mouse model

To generate the prostate-specific *Pten* knockout mice, we crossed Pbsn-Cre male mice Tg (Pbsn-cre) 4Prb/J (Jackson laboratory, Strain: 026662) with Pten floxed mice (B6.129S4-Ptentm1Hwu/J, Strain: 006440). The genotype of the mice was confirmed using the primers provided in the Supplementary Data 1.

## Mouse prostate organoid culture

Eight-week-old wild-type C5B6 and Pten floxed mice were euthanized and the prostate glands were dissected in DMEM 10% FBS. The prostate gland was transferred to a new petri dish with fresh dissecting media and minced with a razor blade. The minced tissue was digested with collagenase on a shaker at 37 °C for 2 h. Tissue chunks were spun down and digested with Trypsin/0.05% EDTA for 5 min at 37 °C. The cell/tissue suspension was passed through a 40 µm nylon mesh filter to get a single-cell suspension. 10,000 cells in 40 µL PrEGM media were mixed with 60 µL Matrigel and seeded to the rim of a 12-well plate, followed by adding 800 µL warm PrEGM in 30 min. In 10 days, both wild-type and Pten floxed P0 organoids were digested with Dispase and Trypsin into single-cell suspension and treated with Adeno-cre for 20 min to induce the knockout in the Pten floxed cells. Ten thousand treated cells in each group were seeded in 12-well plates, images were taken on day 10 and organoid sizes were measured using ImageJ. Genomic DNA and protein lysates were harvested on day 10 for the indicated molecular analysis[86].

## Mouse prostate immunohistochemistry

Mouse prostate tissues were fixed in 4% of formaldehyde for 48 h, and paraffin-embedded through Molecular Pathology and Imaging Core at UPenn. Paraffin-embedded sections from the prostate of Pten floxed control and Pten knockout mice were deparaffinized with 3 changes of xylene for 5 min each. Slides were then rehydrated in 100% alcohol for 10 min, 95% alcohol twice for 10 min each, and 70% alcohol and distilled water for 10 min each. Slides were subjected to citrate-based (pH 6.0) antigen retrieval (Vector, H-3300) at 95 °C for 30 min and followed by blocking endogenous peroxidase activity with 3% $H_2O_2$ for 5 min.

After three times 5 min TBST washing, slides were blocked with blocking buffer (1.25% of goat serum) at room temperature for 1 h. Slides were applied with diluted primary antibody and incubated in a humidified chamber at 4 °C overnight. After three 10 min TBST washes, slides were incubated with secondary antibody (Vector, PK-4001) for 30 min at room temperature the color of the antibody staining was revealed by peroxidase-based detection (Vector, SK4100) and the sections were counterstained with hematoxylin (Millipore Sigma, MHS1). Representative photographs were taken on a Keyence BZ-X Series All-in-one Fluorescence Microscope with 20x and 40x objectives. Images were assessed and quantified in ImageJ.

## Murine prostate tumor xenografts

Eight million LNCaP expressing shNTC, shLCMT (knockdown study) were subcutaneously injected into the right flank or both the dorsal flanks of 5–8-week-old male non-castrated and castrated NCI/NOD SCID/NCr mice (Charles River, strain code: 561, Jackson Laboratory, strain code: 005557) in 50% Matrigel (Corning, 354234). Castration surgery was performed by Charles River before receipt of the mice. The tumor measurement study ended at 40 days, and mice with tumors were sacrificed due to body condition or used for subsequent analysis. Tumor tissue was harvested and was snap frozen in liquid nitrogen for immunoblotting analysis. Mice that did not form tumors at this time continued to be monitored for tumor growth and body weight for first detection tumor analysis, up to 80 days. Eight million LNCaP LacZ or LCMT1 (overexpression study) were subcutaneously injected into the right flank or both the dorsal flanks of 5–8-week-old male non-castrated NCI/NOD SCID/NCr mice (Charles River, strain code: 561, Jackson Laboratory, strain code: 005557). Similarly, $2 \times 10^6$ VCaP cells stably overexpressing LCMT1 along with the control VCaP cells were injected into the dorsal flanks of the non-castrated NOD-SCID mice (Jackson Laboratory, strain code: 005557). The chi-square- or t-test was used to evaluate the association of individual tumor characteristics with engraftment.

For in vivo efficacy studies of DT-061 $2 \times 10^6$ VCaP-EnzaR prostate cancer cells suspended in 80 μL of RPMI 1640 with 50% Matrigel were implanted subcutaneously into the dorsal flanks of the mice. When the tumor volumes reached approximately 80–90 mm³ mice were randomized into three treatment groups viz 15 mg/kg body weight DT-061, 50 mg/kg body weight DT-061 and vehicle-DMA (10%). The treatment was given orally b.i.d for 5 days a week for 3 weeks. In all studies tumors were monitored and recorded by digital caliper twice weekly, and animals were weighed twice weekly. The tumor volumes were estimated using the formula $(\pi/6)$ ($L \times W^2$), where $L$ = length and $W$ = width of tumor. At the end of the treatment regimen the mice were sacrificed, and the tumors were extracted for further analysis.

## Reporting summary

Further information on research design is available in the Nature Portfolio Reporting Summary linked to this article.

# Data availability

All RNA-sequencing and ChIP-sequencing data generated in this study have been deposited in the NCBI Gene Expression Omnibus and are accessible through the GEO Series accession numbers GSE205419 and GSE205348. Source data used in the main figure and supplemental figure panels are also provided. Source data are provided with this paper.

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

## Acknowledgements

The authors thank members of E.O., G.N., and I.A.A lab for comments and suggestions related to the study. Additionally, we would like to thank Prof. Jeffry B. Stock (Department of Molecular Biology, Princeton University) for the Anti amidated-PP2Ac antibody 4D9; Brynne Raines and Rachael Ziskind for technical assistance. This research was supported by grants from the NIH and DoD to I.A.A. (1-R01 CA249210-0 and W81XWH-17-0404). The work of the E.O. lab was funded by service and royalty fees from antibody licensing agreements and the monoclonal antibody service facility of the Medical University of Vienna/Max Perutz Labs.

## Author contributions

R.Rasool, E.O., G.N., and I.A.A. designed the experiments. R.Rasool, C.O., C.K.D., B.K.V., I.E.F, J.Sangodka, A.A., Q.D., E.M-V., S.L., I.M., S.I., J.R., K.P.Z., B.V., C.C. and T.K.S performed the experiments. M.A., R.Rebernicki, and R.N. ran the bioinformatics pipelines and performed data analysis. D.T., R.S., and G.S., performed human TMA staining and analysis. J. Siddiqui, S.W., D.J.T., L.B., M.C., and A.M.C. provided reagents and technical information. I.A.A. wrote the manuscript with input from R.Rasool, E.O., and G.N. All authors read the manuscript and provided comments. E.O., G.N., and I.A.A. supervised the study. I.A.A. conceived the study.

## Competing interests

G.N. has an equity interest in RAPPTA Therapeutics and serves as consultants, along with C.O. E.O. serves as a consultant to Millipore Corporation and RAPPTA Therapeutics and has option interests in RAPPTA Therapeutics. The Medical University of Vienna and the Regents of The University of Michigan on behalf of the authors EO and GN have filed patent WO2021148681A1 on the PP2A methyl-C subunit specific monoclonal antibody 7C10-C5 with the title: Antibodies specifically binding the carboxymethylated catalytic subunit of protein phosphatase 2A. All other authors declare no competing interests.

## Additional information

Reyaz ur Rasool[1], Caitlin M. O'Connor[2,3], Chandan Kanta Das[1], Mohammed Alhusayan[1], Brijesh Kumar Verma[1], Sehbanul Islam[1], Ingrid E. Frohner [4], Qu Deng [1], Erick Mitchell-Velasquez[1], Jaya Sangodkar[2,3], Aqila Ahmed[2,3], Sarah Linauer[4], Ingrid Mudrak[4], Jessica Rainey [1], Kaitlin P. Zawacki[2,3], Tahra K. Suhan[2,3], Catherine G. Callahan[2,3], Ryan Rebernick[5,6], Ramakrishnan Natesan[1], Javed Siddiqui[5,6], Guido Sauter[7], Dafydd Thomas[6], Shaomeng Wang[8], Derek J. Taylor[9], Ronald Simon[7], Marcin Cieslik[5,6], Arul M. Chinnaiyan [5,6], Luca Busino [1], Egon Ogris[4] ✉, Goutham Narla [2,3] ✉ & Irfan A. Asangani [1,10,11] ✉

[1]Department of Cancer Biology, Perelman School of Medicine, University of Pennsylvania, 421 Curie Boulevard, BRBII/III, Philadelphia, PA 19104, USA. [2]Division of Genetic Medicine, Department of Internal Medicine, University of Michigan, Ann Arbor, MI 48105, USA. [3]Rogel Cancer Center, University of Michigan, Ann Arbor, MI 48109, USA. [4]Center for Medical Biochemistry, Max Perutz Labs, Medical University of Vienna, Dr. Bohr-Gasse 9/2, Vienna 1030, Austria. [5]Department of Pathology and Rogel Cancer Center, University of Michigan Medical School, Ann Arbor, MI, USA. [6]Michigan Center for Translational Pathology, University of Michigan, Ann Arbor, MI, USA. [7]Institute of Pathology, University Medical Center Hamburg-Eppendorf, 20246 Hamburg, Germany. [8]Departments of Internal Medicine, Pharmacology, and Medicinal Chemistry, University of Michigan, Ann Arbor, MI, USA. [9]Department of Biochemistry Case Western Reserve University School of Medicine, Cleveland, OH 44106, USA. [10]Abramson Family Cancer Research Institute, Perelman School of Medicine, University of Pennsylvania, Philadelphia, PA, USA. [11]Epigenetics Institute, Perelman School of Medicine, University of Pennsylvania, Philadelphia, PA, USA. ✉e-mail: egon.ogris@meduniwien.ac.at; gnarla@med.umich.edu; asangani@upenn.edu

