## [Peer Review File · Nature Communications]

Loss of LCMT1 and biased protein phosphatase 2A heterotrimerization drive prostate cancer progression and therapy resistanceREVIEWER COMMENTS

Reviewer #1 (Remarks to the Author):

Functional Interplay between protein kinases and phosphatases in regulating protein phosphorylation is fundamental to numerous biological processes and diseases. Serine/threonine protein phosphatase 2A (PP2A) has been reported as a tumor suppressor by dephosphorylating target proteins through LCMT1-dependent AB56 α C subunits assembly. However, the molecular basis of PP2A dysregulation in advanced prostate cancer (PCa) remains uncharacterized. In this study, Rasool et al. showed that loss of methyl-PP2A-C was associated with shorter biochemical recurrence survival and unfavorable clinical parameters of prostate cancer progression. Depletion of LCMT1 disrupted the PP2A heterotrimer assembly and increased phosphorylation of AR and MED1, resulting in activated AR signaling and enhanced castration-independent tumor growth. LCMT1-mediated dephosphorylation of AR/MED1 was achieved through binding of the PP2A B56 α regulatory subunit to two conserved small linear motifs (SLiMs) on MED1. Mechanistically, LCMT1 was regulated by mTOR ribosomal protein S6 kinase 1 (S6K1)-mediated phosphorylation-induced degradation requiring the β -TRCP E3 ubiquitin ligase. Lastly, stabilization of LCMT1 by a small molecule activator of phosphatase (SMAP) or treatment with an AR degrader resulted in suppression of AR signaling and dramatic tumor growth inhibition in advanced PCa xenografts. Overall, this is a very comprehensive and detailed study, and the manuscript is well-written. A few comments below need to be addressed to improve the manuscript.

1. In figure 5, authors showed S6K1 phosphorylated LCMT1 to decrease its abundance. Did S6K1 directly interact with LCMT1?
2. β -TrCP was identified as an E3 ubiquitin ligase targeting phosphorylated LCMT1 for degradation. Both in vitro and in vivo ubiquitination assays should be added to further support its roles in regulating LCMT1 stability in figure 5.
3. In Extended Data Fig. 7c, the WB analysis of AR, PTEN, pAKT, AKT, pS6 and S6 should be run in the same gel.
4. In Extended Data Fig. 7f, Pten, Myc, pS6 IHC staining signals in fl/fl mice prostate seem to have no significant differences from those in pten^{-/-} mice prostate.
5. In Extended Data Fig. 11, as none of normal prostate cells/models were employed in the study, the "normal" should be corrected to "primary" in the schematic.
6. The resolution of Extended Data Fig.2b, Fig.3c, Fig.5f, Fig. 8d and Fig.8e was poor and needed to be improved.

Reviewer #2 (Remarks to the Author):

The manuscript titled "Loss of LCMT1 and biased protein phosphatase 2A heterotrimerization drive prostate cancer progression and therapy resistance" demonstrate that the loss of LCMT1 is a significant determinant of AR-addicted PCa, and supports the therapeutic potential of AR degraders or selective small molecules modulators of PP2A for prostate cancer treatment. PP2A reactivation by SMAPs such as DT-061 can restore LCMT1 through a feedforward mechanism involving inhibition of the PI3K pathway and reverse AR addiction in second-generation anti-androgen refractory CRPC.

However, there are some minor questions:

1. In The mouse model for prostate cancer, about five months will develop HGPIN. How many mice were used in this study, and what proportion will develop HGPIN in 4-5 months? What other mice's pathological results?
2. In the discussion part, could we add more limitations or following step research of this

study?

3. VCaP cell line was used for comparison to LNCaP for AR expression; do we need a negative control like PC-3 cell lines for suggestions?
4. Figure 2E, better add one more shLCMT1 for reference.
5. Figure 5E why do we need to use HEK293-T other than different prostate cancer cells?

The manuscript explores the spread aspects of the pathways and draws a picture of LCMT1's role in prostate cancer progression and resistance.

3. VCaP cell line was used for comparison to LNCaP for AR-expression, do we need a negative control like PC-3 cell lines for suggestions?
4. Figure 2E, better add one more shLCMT1 for reference.
5. Figure 5E why we need to use HEK293-T other than different prostate cancer cells?

The manuscript explore spread aspects of the pathways and draw a picture on LCMT1 role of prostate cancer progression and resistance.

Reviewer #3 (Remarks to the Author):

This is a rigorously conducted mechanistic study that reports on the role of leucine carboxy methyltransferase (LCMT1) activity on the regulation of the PP2A/B56 α holoenzyme as a negative feedback suppressor of Androgen Receptor (AR) Signaling in AR-addicted prostate cancer (PCa). The authors show that loss of PP2A/C carboxymethylation (mediated by LCMT1 downregulation) in prostate tumors is associated with PCa recurrence and metastasis. The authors find that the LCMT1 protein is degraded by the proteasome following S6K1-mediated phosphorylation of residues at its N-terminus that allow recruitment of the ubiquitin ligase β -TRCP. LCMT1 downregulation results in a relative decrease on PP2A/B56 α and PP2A/B55 α holoenzymes and PP2A/B56 α is responsible for the specific dephosphorylation of MED1 and AR, through two short linear motifs (SLIM) present in MED1. Therefore, upregulation of AKT/mTOR/S6K1 signaling, which is common in PCa, downregulates LCMT1, resulting in potent attenuation of PP2A/B56 α , which in turn results in increased AR/MED1 retention in chromatin. Of note, in PCa cell lines resistant to anti-AR drugs, AKT/S6K1 signaling is upregulated leading to dramatic LCMT1 downregulation. The authors then show that a small molecule, previously developed to promote PP2A/B56 holoenzyme assembly, results in inhibition of S6K1, and upregulation of LCMT1 in enzalutamide-resistant PCa cells reducing AR and MED1 phosphorylation, chromatin binding, AR enhancer activity and AR-dependent gene expression and xenograft tumor growth.

In this manuscript, the authors report on several novel mechanistic steps linking LCMT1 regulation of a specific PP2A/B56 holoenzyme to AR signaling identifying all critical players using loss and gain of function experiments, appropriate controls and multiple cell lines. The reproducibility of the findings in the multiple cell lines is remarkable and attest to the generality and importance of these signaling circuits. The mechanistic work unveils the key steps leading to LCMT1 downregulation in tumors. Another important contribution of this work is the characterization of the mouse mab 7C10-C5, which specifically recognizes PP2A/C methyl-L309, as downregulation of this posttranslational modification could serve as a prognostic biomarker for biochemical recurrence and metastasis resistance, which could be potentially actionable.

The manuscript is concisely written and the data support the authors conclusions. The work is impactful given its significance in prostate cancer but also of general biological interest, as it unveils a novel mechanism for the control of AR-signaling by PP2A at the chromatin level. We work will be of interest to a wide audience. Also, the extended data should be of interest to those wanting to obtain additional details needed for follow up work.

There are only a couple of issues that need to be addressed in this manuscript:

The first is whether the similarly downregulated PP2A/B55 α complexes play any role in this pathway and ARSI resistance, as PP2A/B55 α has been implicated by different labs in negative regulation of AKT through dephosphorylation of its activating sites. I do not think that this needs to be explored further in this manuscript, but it should be concisely discussed.

The second, is that the manuscript does not show data demonstrating that DT-061 increases PP2A/B56 α complexes in the relevant PCa cell lines, and given the alternative mechanism of action proposed for DT-061 in Vit et al., 2022 (The EMBO Journal (2022)41:e110611), it would be important to show IP data performed with PCa-EnzaR cells showing that DT-061 stabilizes these complexes.

We are extremely thankful to all the reviewers for their positive and constructive comments on our manuscript. We have performed several additional experiments to address the questions from the reviewers. The new data and discussions have been included in the revised manuscript (highlighted with gray background). Below we present a point-by-point rebuttal of the reviewers' comments.

Reviewer #1 (Reviewer Comments to the Author):

Functional Interplay between protein kinases and phosphatases in regulating protein phosphorylation is fundamental to numerous biological processes and diseases. Serine/threonine protein phosphatase 2A (PP2A) has been reported as a tumor suppressor by dephosphorylating target proteins through LCMT1-dependent AB56 α C subunits assembly. However, the molecular basis of PP2A dysregulation in advanced prostate cancer (PCa) remains uncharacterized. In this study, Rasool et al. showed that loss of methyl-PP2A-C was associated with shorter biochemical recurrence survival and unfavorable clinical parameters of prostate cancer progression. Depletion of LCMT1 disrupted the PP2A heterotrimers assembly and increased phosphorylation of AR and MED1, resulting in activated AR signaling and enhanced castration-independent tumor growth. LCMT1-mediated dephosphorylation of AR/MED1 was achieved through binding of the PP2A B56 α regulatory subunit to two conserved small linear motifs (SLiMs) on MED1. Mechanistically, LCMT1 was regulated by mTOR ribosomal protein S6 kinase 1 (S6K1)-mediated phosphorylation-induced degradation requiring the β -TRCP E3 ubiquitin ligase. Lastly, stabilization of LCMT1 by a small molecule activator of phosphatase (SMAP) or treatment with an AR degrader resulted in suppression of AR signaling and dramatic tumor growth inhibition in advanced PCa xenografts. Overall, this is a very comprehensive and detailed study, and the manuscript is well-written. A few comments below need to be addressed to improve the manuscript.

Response: We thank the reviewer for summarizing our work and appreciate the positive comments. We have performed additional experiments that establish the central role of LCMT1/PP2A/-S6K1 axis in AR addiction in advanced prostate cancer.

1. In figure 5, authors showed S6K1 phosphorylated LCMT1 to decrease its abundance. Did S6K1 directly interact with LCMT1?

Response: To further consolidate our observations that S6K1 targets LCMT1 for phosphorylation directly, we performed an immunoprecipitation assay, where we co-expressed V5-LCMT1 and Flag-S6K1 constructs in HEK293-T cell. A reciprocal pull-down with V5 or Flag antibody followed by immunoblotting demonstrated an interaction between LCMT1 and S6K1 (**new Extended Data Fig.6g**). The new results have been described accordingly in the text.

2. β -TrCP was identified as an E3 ubiquitin ligase targeting phosphorylated LCMT1 for degradation. Both *in vitro* and *in vivo* ubiquitination assays should be added to further support its roles in regulating LCMT1 stability in figure 5.

Response: Following the reviewer's suggestion, we have performed both *in vivo* and *in vitro* ubiquitination studies. The experimental results further supported our studies showing a key role of E3 ubiquitin ligase β -TrCP in the degradation of LCMT1. For *in vivo* ubiquitination assay, we first checked whether the knockdown of β -TrCP results in reduced LCMT1 polyubiquitination. A marked reduction in the polyubiquitinated LCMT1 was evident upon β -TrCP depletion compared to siNT transfected HEK293-T cells (**new Extended Data Fig. 6i**). Next, for *in vivo* ubiquitination assay, HEK293T cells were transfected

with His-ubiquitin, along with either FLAG-LCMT1, HA- β -TrCP, or both, to demonstrate direct ubiquitination of LCMT1 by β -TrCP in vivo. Pull-down of ubiquitinated proteins using Ni-NTA beads, followed by immunoblotting, revealed increased levels of polyubiquitinated FLAG-LCMT1 upon β -TrCP overexpression (new Extended Data Fig. 6j). Finally, we performed in vitro ubiquitination assays. In addition to LCMT1 and S6K1, the reaction mixtures contained known cofactors (E1, E2, ubiquitin, and ATP), and β -TrCP protein. High-molecular weight species (polyubiquitinated) of LCMT1 were detected only when β -TrCP was present in the reaction mix (new Extended Data Fig. 6k). Together, these data demonstrate β -TrCP as a key cullin1 Ring Ligase receptor responsible for LCMT1 degradation. The new in vivo and in vitro ubiquitination results have been described accordingly in the revised text.

3. In Extended Data Fig. 7c, the WB analysis of AR, PTEN, pAKT, AKT, pS6 and S6 should be run in the same gel.

Response: We appreciate and understand the reviewer's concern. The WB data in Extended Data Fig. 7c for AR, PTEN, pAKT, AKT, pS6, and S6 were indeed from the same blot (please see the original unspliced blots in the adjacent rebuttal figure). However, we spliced out the lanes irrelevant to this study and presented only the fl/fl and pten -/- lanes. For further clarity, we have revised the sentence with "spliced lanes are from the same gel" in the figure legend.

4. In Extended Data Fig. 7f, Pten, Myc, pS6 IHC staining signals in fl/fl mice prostate seem to have no significant differences from those in pten -/- mice prostate.

Response: Following the reviewer's concern, we have replaced the representative IHC staining figures with the new images with a higher resolution/magnification that clearly show differences in pten -/- condition.

5. In Extended Data Fig. 11, as none of normal prostate cells/models were employed in the study, the "normal" should be corrected to "primary" in the schematic.

Response: We appreciate the reviewer's comment. However, the schematic represents a general mechanism of LCMT1 regulation through the PI3K/S6K pathway and how its deregulation can lead to pathological conditions like cancer. The data from HEK293T cells (human embryonic kidney cells) – Fig. 5e,h,i,m and Extended Data Fig.6f, and i; NIH3T3 (mouse fibroblast cells) - Extended Data Fig.7e,f; and normal mouse prostate (flox) - Extended Data Fig.7, support the significance of this signaling axis in maintaining normal cell and tissue homeostasis, which is disrupted in malignancies, especially prostate cancer.

6. The resolution of Extended Data Fig.2b, Fig.3c, Fig.5f, Fig. 8d and Fig.8e was poor and needed to be improved.

Response: We appreciate the reviewer's comment. The poor resolution of the figures is due to file size being reduced during manuscript submission. The higher resolution original figures in AI and pdf formats are being submitted in this revision.

Reviewer #2 (Reviewer Comments to the Author):

The manuscript titled "Loss of LCMT1 and biased protein phosphatase 2A heterotrimerization drive prostate cancer progression and therapy resistance" demonstrate that the loss of LCMT1 is a significant determinant of AR-addicted PCa and supports the therapeutic potential of AR degraders or selective small molecules modulators of PP2A for prostate cancer treatment. PP2A reactivation by SMAPs such as DT-061 can restore LCMT1 through a feedforward mechanism involving inhibition of the PI3K pathway and reverse AR addiction in second-generation anti-androgen refractory CRPC.

Response: We thank the reviewer for summarizing our work.

1. *In the mouse model for prostate cancer, about five months will develop HGPIN. How many mice were used in this study, and what proportion will develop HGPIN in 4-5 months? What other mice's pathological results?*

Response: We thank the reviewer for this comment. A total of nine (six-month-old) *Pten* null mice were used in the study, all of them developed HGPIN and/or florid lesions and ductal carcinoma *in situ*. Seven flox control mice were used, and as expected no significant pathological changes were observed. The description has also been inserted into the main text.

2. *In the discussion part, could we add more limitations or following step research of this study?*

Response: Following the suggestion, we have slightly revised the discussion by including sentences on limitations and future directions.

3. *VCaP cell line was used for comparison to LNCaP for AR expression; do we need a negative control like PC-3 cell lines for suggestions?*

Response: We thank the reviewer for the comment. Our study is designed to follow the molecular events associated with androgen receptor signaling and its reactivation in anti-androgen refractory state. PC3 being an androgen receptor-independent cell line shows no such signaling and does not respond to anti-androgens like enzalutamide.

4. *Figure 2E, better add one more shLCMT1 for reference.*

Response: To further confirm the effect of shRNA mediated LCMT1 knock-down on AR signaling under androgen deprivation, we have now included data for two more cell line models viz shLCMT1- VCaP and -LAPC4. As observed in the LNCaP shLCMT1 cells (Fig. 2E), continued transcription of AR target genes was evident in both VCaP and LAPC4 shLCMT1 cells under androgen deprivation (charcoal stripped FBS condition (**new Extended Data Fig.3c**)).

5. *Figure 5E why do we need to use HEK293-T other than different prostate cancer cells?*

Response: We thank the reviewer for their comment. To demonstrate a conserved mechanism for LCMT1 regulation through the PI3K/AKT/S6K1 pathway, experiments were conducted in both HEK293T and PCa cells. It was critical to validate this regulatory pathway in HEK293T cells, as this cell line was used, for ease of transfection, in all downstream signaling, mapping, and ubiquitination experiments presented in Fig. 5f-m and Extended Data Fig. 6f-k. Additionally, the generalizability of these findings to non-prostate cancer models could extend the impact of the work to other tumor types.

Reviewer #3 (Reviewer Comments to the Author):

This is a rigorously conducted mechanistic study that reports on the role of leucine carboxy methyltransferase (LCMT1) activity on the regulation of the PP2A/B56 α holoenzyme as a negative feedback suppressor of Androgen Receptor (AR) Signaling in AR-addicted prostate cancer (PCa). The authors show that loss of PP2A/C carboxymethylation (mediated by LCMT1 downregulation) in prostate tumors is associated with PCa recurrence and metastasis. The authors find that the LCMT1 protein is degraded by the proteasome following S6K1-mediated phosphorylation of residues at its N-terminus that allow recruitment of the ubiquitin ligase β -TRCP. LCMT1 downregulation results in a relative decrease on PP2A/B56 α and PP2A/B55 α holoenzymes and PP2A/B56 α is responsible for the specific dephosphorylation of MED1 and AR, through two short linear motifs (SLiM) present in MED1. Therefore, upregulation of AKT/mTOR/S6K1 signaling, which is common in PCa, downregulates LCMT1, resulting in potent attenuation of PP2A/B56 α , which in turn results in increased AR/MED1 retention in chromatin. Of note, in PCa cell lines resistant to anti-AR drugs, AKT/S6K1 signaling is upregulated leading to dramatic LCMT1 downregulation. The authors then show that a small molecule, previously developed to promote PP2A/B56 holoenzyme assembly, results in inhibition of S6K1, and upregulation of LCMT1 in enzalutamide-resistant PCa cells reducing AR and MED1 phosphorylation, chromatin binding, AR enhancer activity and AR-dependent gene expression and xenograft tumor growth.

In this manuscript, the authors report on several novel mechanistic steps linking LCMT1 regulation of a specific PP2A/B56 holoenzyme to AR signaling identifying all critical players using loss and gain of function experiments, appropriate controls, and multiple cell lines. The reproducibility of the findings in the multiple cell lines is remarkable and attests to the generality and importance of these signaling circuits. The mechanistic work unveils the key steps leading to LCMT1 downregulation in tumors. Another important contribution of this work is the characterization of the mouse mab 7C10-C5, which specifically recognizes PP2A/C methyl-L309, as downregulation of this posttranslational modification could serve as a prognostic biomarker for biochemical recurrence and metastasis resistance, which could be potentially actionable. The manuscript is concisely written, and the data support the authors conclusions. The work is impactful given its significance in prostate cancer but also of general biological interest, as it unveils a novel mechanism for the control of AR-signaling by PP2A at the chromatin level. The work will be of interest to a wide audience. Also, the extended data should be of interest to those wanting to obtain additional details needed for follow-up work. There are only a couple of issues that need to be addressed in this manuscript:

Response: We are extremely thankful to the reviewer for appreciating our work and for providing encouraging feedback.

1. The first is whether the similarly downregulated PP2A/B55 α complexes play any role in this pathway and ARSI resistance, as PP2A/B55 α has been implicated by different labs in negative regulation of AKT

through dephosphorylation of its activating sites. I do not think that this needs to be explored further in this manuscript, but it should be concisely discussed.

Response: We thank the reviewer for raising this critical point. We acknowledge the potential role of PP2A/B55 α complexes in the AKT/S6K1/LCMT1 pathway and resistance to ARSI is an interesting topic, however, as the reviewer rightly pointed out, it is beyond the scope of the current study. Following the reviewer's suggestion, we have added a brief discussion alluding to the potential role of PP2A/B55 α complexes in regulating the AKT pathway and AR signaling in the context of ARSI refractory state.

2. The second, is that the manuscript does not show data demonstrating that DT-061 increases PP2A/B56 α complexes in the relevant PCa cell lines, and given the alternative mechanism of action proposed for DT-061 in Vit et al., 2022 (The EMBO Journal (2022)41:e110611), it would be important to show IP data performed with PCa-EnzaR cells showing that DT-061 stabilizes these complexes.

Response: We thank the reviewer for this critical comment. To demonstrate the effect of DT-061 in increasing the stability of PP2A/B56 α complexes in PCa cells, we treated LNCaP cells stably expressing PP2A-V5 with 10 μ M DT-061 for 12h, followed by V5-IP. Increased AB56Cme heterotrimers was evident in DT-061 treated cells compared to control. Next, we investigated the effect of DT-061 on PP2A heterotrimer formation and target dephosphorylation in xenograft tumors derived from VCaP EnzaR cells in vivo. Compared to the vehicle-treated mice, protein lysates from VCaP EnzaR residual tumors of DT-061-treated mice showed a decrease in phosphorylated AR, MED1, and S6K1 levels and an increased LCMT1 level (**new Extended Data Fig. 10h**). Furthermore, the B56-alpha pull-down followed by immunoblotting revealed higher levels of PP2A heterotrimers (AB56 α Cme) in the DT-061-treated tumor lysates (50mg/kg) than in the vehicle group (**new Extended Data Fig.10i**). The cell line and in vivo tumor lysate data clearly demonstrate the ability of DT-061 to stabilize specific PP2A heterotrimers and target dephosphorylation.

REVIEWERS' COMMENTS

Reviewer #1 (Remarks to the Author):

The authors have satisfactorily addressed my concerns. I also note that they have also taken into account the concerns raised by reviewer 2, specifically regarding the prostate cancer mouse model and the selection of cell lines. The manuscript is ready for publication.

Reviewer #3 (Remarks to the Author):

The authors have addressed my concerns. I also think they made a great job addressing the concerns from the other reviewers. Therefore, I am completely satisfied with the revised version of this manuscript and feel it is acceptable for publication in Nature Communications.